# Towards sector-based attribution using intra-city variations in satellite-based emission ratios between $CO_2$ and CO

**Dien Wu**[1], **Junjie Liu**[2,1], **Paul O. Wennberg**[1,3], **Paul I. Palmer**[2,4], **Robert R. Nelson**[2], **Matthäus Kiel**[2], **and Annmarie Eldering**[2]

[1]Division of Geological and Planetary Sciences, California Institute of Technology, Pasadena, USA
[2]Jet Propulsion Laboratory, California Institute of Technology, Pasadena, USA
[3]Division of Engineering and Applied Science, California Institute of Technology, Pasadena, USA
[4]School of GeoSciences, University of Edinburgh, Edinburgh, UK

**Correspondence:** Dien Wu (dienwu@caltech.edu)

**Abstract.** Carbon dioxide ($CO_2$) and air pollutants such as carbon monoxide (CO) are co-emitted by many combustion sources. Previous efforts have combined satellite-based observations of multiple tracers to calculate their emission ratio (ER) for inferring combustion efficiency at the regional to city scale. Very few studies have focused on combustion efficiency at the sub-city scale or related it to emission sectors using space-based observations. Several factors are important for interpreting and deriving spatially resolved ERs from asynchronous satellite measurements, including (1) variations in meteorological conditions given the mismatch in satellite overpass times, (2) differences in vertical sensitivity of the retrievals (i.e., averaging kernel profiles), (3) interferences from the biosphere and biomass burning, and (4) the mismatch in the daytime variations of CO and $CO_2$ emissions. In this study, we extended an established emission estimate approach to arrive at spatially resolved ERs based on retrieved column-averaged $CO_2$ ($XCO_2$) from the Snapshot Area Mapping (SAM) mode of the Orbiting Carbon Observatory-3 (OCO-3) and column-averaged CO from the TROPOspheric Monitoring Instrument (TROPOMI).

To evaluate the influences of the confounding factors listed above and further attribute intra-urban variations in ERs to certain sectors, we leveraged a Lagrangian atmospheric transport model with an urban land cover classification dataset and reported $ER_{CO}$ values from the sounding level to the overpass and city level. We found that the differences in overpass times and averaging kernels between OCO and TROPOMI strongly affect the estimated spatially resolved $ER_{CO}$. Specifically, a time difference of $> 3\,h$ typically led to dramatic changes in wind directions and urban plume shapes, thereby making the calculation of accurate sounding-specific $ER_{CO}$ difficult. After removing such cases from consideration and applying a simple plume shift method when necessary to account for changes in wind direction and speed, we discovered significant contrasts in combustion efficiencies between (1) two megacities versus two industry-oriented cities and (2) different regions within a city, based on six nearly coincident overpasses per city. Results suggest that the $ER_{CO}$ impacted by heavy industry in Los Angeles is slightly lower than the overall city-wide value ($< 10\,\text{ppb-CO/ppm-}CO_2$ TS1). In contrast, the $ER_{CO}$ related to heavy industry in Shanghai is much higher than Shanghai's city mean and more aligned with the city means of two selected industry-oriented cities in China (approaching $20\,\text{ppb-CO/ppm-}CO_2$ TS2). Although investigations based on a larger number of satellite overpasses are needed, our unique approach (i.e., without using sector-specific information from emission inventories) provides new insights into assessing combustion efficiency within a city from future satellite missions, such as those that will map column $CO_2$ and CO concentrations simultaneously with high spatiotemporal resolutions.

## 1 Introduction

Home to more than half of the total global population, urban areas have been expanding, especially in Asia and Africa, which had urbanization rates of 1.3 % and 1.1 % $yr^{-1}$, re-
5 spectively, between 2015 and 2020 (United Nations et al., 2020) TS3. Urban regions are also responsible for a significant amount of anthropogenic emissions of greenhouse gases (GHG) and air pollutants into the atmosphere, including carbon dioxide ($CO_2$), methane, carbon monoxide (CO), and ni-
10 trogen oxides (Duncan et al., 2016; Lin et al., 2018; Super et al., 2017; Plant et al., 2019). Satellite observations have become indispensable for monitoring the abundances of several atmospheric species in a globally consistent manner (Yokota et al., 2009; Crisp et al., 2012; Veefkind et al., 2012). For
example, carbon-monitoring satellites such as Orbiting Carbon Observatory-2 (OCO-2, Crisp et al., 2012) have made the quantification of city-scale $CO_2$ emissions and emission trends possible (e.g., Hedelius et al., 2018; Ye et al., 2020; Wu et al., 2020; Shekhar et al., 2020; Lei et al., 2021). Quan-
tifying the spatial gradient of atmospheric concentrations and relating this gradient to emissions within the city domain has become the next critical yet challenging task. Understanding such spatial heterogeneity in emissions and the environmental consequences of it can support better decisions in urban
planning and the pinpointing of hotspots for emission mitigation.

   Given the co-benefit of GHG reduction and improved air quality at various scales (Zhang et al., 2017), controlling the consumption of fossil fuels altogether is the key. The efficien-
30 cies associated with various combustion activities are linked to their underlying processes and conditions (e.g., oxygen-to-fuel ratio and temperature). For example, the amount of $CO_2$ emitted from coal-fired power plants varies with thermal and pressure conditions, the type of fuel consumed, the technol-
35 ogy deployed, and the service duration of power plants (Yuan and Smith, 2011). A CE1 modern power generation plant that uses scrubbing technology is often regarded as a CE2 "clean" emitter, leading to minimal CO and $NO_x$ enhancement (Lindenmaier et al., 2014). The commonly used approach when
estimating combustion efficiency is to combine atmospheric observations of multiple trace gases and report the ratio of the total or excess measured concentrations (above a defined background value) of the tracers (Silva and Arellano, 2017; Reuter et al., 2019; Park et al., 2021). Such a tracer-to-
tracer ratio calculation has the benefit that errors in describing the atmospheric transport that carries tracers to the measurement site can be canceled. A few notable studies have further utilized derived emission ratios (ERs) from ground or airborne measurements to infer sector-specific emission sig-
nals (Wennberg et al., 2012; Lindenmaier et al., 2014; Nathan et al., 2018; Tang et al., 2020).

CO and $NO_x$ often serve as tracers for anthropogenic $CO_2$ as they arise from similar sources (e.g., Palmer et al., 2006; Wunch et al., 2009; Hedelius et al., 2018). Analyzing re-
55 motely sensed $NO_x$ plumes with relatively short lifetimes can help identify local fossil fuel $CO_2$ (FFCO$_2$) sources that would otherwise be difficult to detect (Reuter et al., 2019; Fujinawa et al., 2021). At the same time, such reactivity requires that chemical transformations are accurately accounted for
and complicates the interpretation of emission signals or ERs from $NO_x$ observations (Lama et al., 2020; Hakkarainen et al., 2021). Given its much longer lifetime, CO is much easier to interpret and more likely to be found during incomplete combustion. The emission ratio of CO to $CO_2$ (ER$_{CO}$)
is usually estimated from sparse ground-based measurements within a city (Bares et al., 2018; Chandra et al., 2016; Lindenmaier et al., 2014) and from satellites at the city scale (Park et al., 2021; Silva and Arellano, 2017). Sector-specific activities and ER$_{CO}$ values such as those from the traffic sec-
tor have been analyzed in limited but valuable tunnel studies (Ammoura et al., 2014; Bradley et al., 2000; Popa et al., 2014).

   We performed a literature search for ER$_{CO}$ values derived from observations (Appendix A) and the values are summa-
75 rized in Fig. 1. The combustion efficiency fluctuates (1) over time (e.g., Turnbull et al., 2011b), likely due to technological improvements, and (2) between sub-sectors, e.g., gasoline vs. diesel vehicles or moving vs. congestion traffic (Westerdahl et al., 2009; Popa et al., 2014). Despite differences in mea-
80 surement platforms and analytical approaches, the observed urban-integrated ER$_{CO}$ values, especially those in Europe and the United States, are well constrained within the range of 4 to 15 ppb ppm$^{-1}$ (Fig. 1b). ER$_{CO}$ values for biomass burning and shipping sectors are estimated based on fuel-
85 specific emission factors, i.e., ER$_{CO}$ ($= EF_{CO}/EF_{CO_2}$) with proper unit conversions, where each emission factor $EF_X$ indicates the emission of gas $X$ per kg of fuel burned.

   When estimating fossil fuel emissions from a bottom-up perspective, most inventories rely on activity data and may
involve prior knowledge of emission factors (Gurney et al., 2019; Solazzo et al., 2021). One notable example is Hestia, a high-resolution inventory for the US that estimates $CO_2$ emissions of non-point sources based on CO emissions from the National Emission Inventory and EFs and carefully eval-
uates the adopted EFs (Gurney et al., 2019). However, when constructing emission inventories across regions and nations, the large variability in ERs across combustion processes, sectors, years, and regions (as seen in Fig. 1a) makes the choice of EFs extremely challenging. Accurate bottom-up emission
estimates require accurate activity data and $EF_X$ values that naturally vary with the combustion conditions (e.g., temperature, fuel load, oxygen level) and are generally not well known, especially over data-scarce regions. To our knowledge, only a few global inventories, such as the Emissions

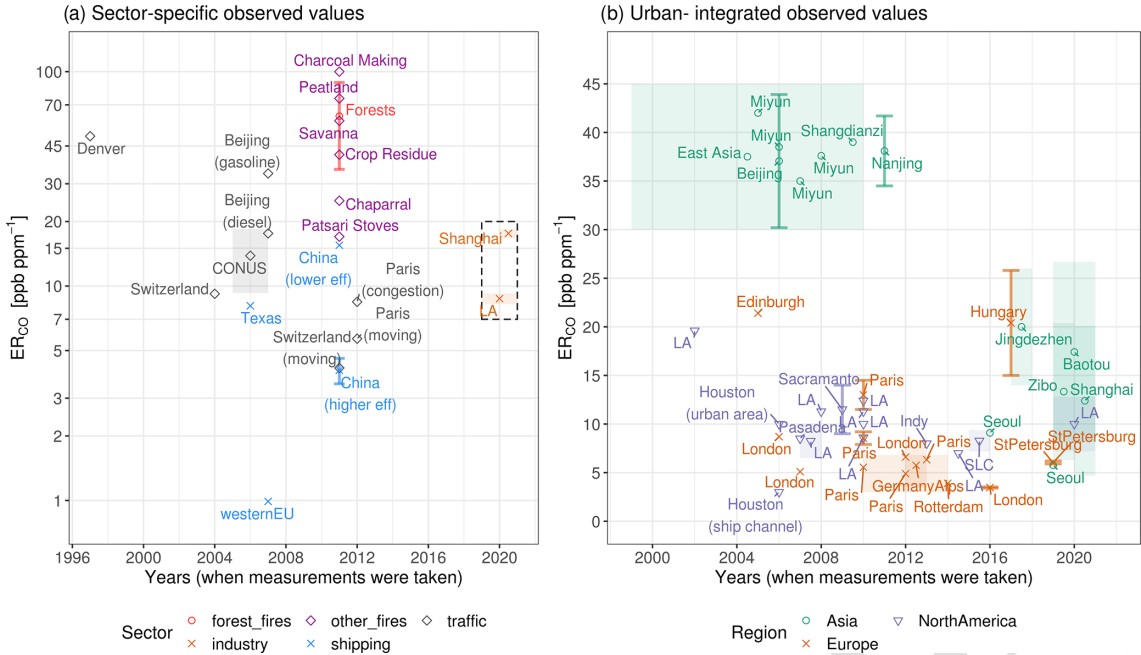

**Figure 1.** $ER_{CO}$ [ppb ppm$^{-1}$] values associated with specific processes **(a)** and $ER_{CO}$ values integrated over the entire city/region **(b)**, as summarized from previous studies. The $x$ axis indicates the times at which these estimates were made, except for Akagi et al. (2011); in the latter case, 2011 was chosen for the $x$ axis since the paper was published in 2011. The error bars represent the uncertainties in the estimated ERs and the shaded rectangles indicate ranges of ERs over multiple years. Paper citations are omitted from the figure but are included in Appendix A. ERs related to the biomass burning and shipping sectors are derived using $EF_{CO}$ and $EF_{CO_2}$. The range of overpass-specific $ER_{CO}$ estimates for Shanghai, LA, Baotou, and Zibo derived from our study are also shown in the figure as a dashed black box.

Database for Global Atmospheric Research (EDGAR, So-lazzo et al., 2021), offer global anthropogenic CO and $CO_2$ emissions. Considering the challenge involved in approximating ERs, certain knowledge derived from atmospheric observations may (1) complement inventory-based ERs (e.g., the CO : NO$_x$ ratio in Lama et al., 2020) and (2) facilitate emission constraints for a desired gas with relatively large uncertainties (Wunch et al., 2009; Palmer et al., 2006; Wang et al., 2009; Brioude et al., 2012; Nathan et al., 2018). Such prior achievements motivate us to examine ERs using satellite observations of multiple tracers.

Most existing studies have focused on quantifying an integrated ER for a whole city or region. We take a step forward, zooming into an urban area and leveraging spatially resolved satellite observations. Intra-city variations in the satellite-based concentration of a specific air pollutant such as NO$_x$ have been analyzed and linked to societal inequalities regarding income and educational attainment (Demetillo et al., 2021; Kerr et al., 2021). Yet, no one has attempted to study the intra-urban gradient in combustion efficiency from space and to relate this gradient to a specific combustion sector. This is now possible by virtue of Orbiting Carbon Observatory-3 (OCO-3, mounted on the International Space Station), which can sample a city landscape during the Snapshot Area Mapping (SAM) mode (Eldering et al., 2019; Taylor et al., 2020; Kiel et al., 2021). In an effort to arrive at

spatially varying ERs from sensors with asynchronous orbits, we must account for several factors that have not been thoroughly investigated. These include (a) variations in meteorological conditions and emission patterns during different overpass times, (b) discrepancies in the vertical sensitivities of the retrievals (i.e., averaging kernel (AK) profiles), and (c) interference from non-anthropogenic sources and sinks, especially from the biosphere.

In this study, we explore the spatial distribution of $ER_{CO}$ within four urban areas, mainly using $XCO_2$ observations from OCO-3 and XCO observations from the TROPO-spheric Monitoring Instrument onboard the Sentinel-5 Precursor (TROPOMI; Veefkind et al., 2012). To avoid relying on prior sector-specific information on $ER_{CO}$ from emission inventories, we adopt the urban land cover data from the high-resolution World Urban Database and Access Portal Tools (WUDAPT; Ching et al., 2018). WUDAPT offers the so-called local climate zone (LCZ) that considers the building structure/spacing along with the vegetation coverage (Stewart and Oke, 2012), which shed light on the urban infrastructure.

Our work seeks to answer the following two questions:

1. Is it possible to accurately quantify the spatially resolved $ER_{CO}$ from asynchronous satellite measurements?

2. Can the combustion efficiency for a given sector be determined without using prior emission inventories?

In Sect. 2, we describe the satellite data and methodology used for obtaining emissions, the $ER_{CO}$, and associated uncertainties. In Sect. 3, we show intra-city variations in $ER_{CO}$ (including the $ER_{CO}$ tied to heavy industry in a megacity) and how multiple factors may interfere when deriving $ER_{CO}$. In Sect. 4, we discuss the implications and limitations of this analysis.

## 2   Data and methodology

We target two types of cities: (1) industry- and energy-oriented cities (Baotou, China and Zibo, China) and (2) megacities with more diverse emission sectors (Shanghai, China and Los Angeles, USA). The four cities are selected considering the amount and quality of $XCO_2$ data from OCO-3 SAMs and TROPOMI XCO data. The two industry- and energy-oriented cities in China are selected given their large amount of metal production plants for aluminum or iron and steel (Global Energy Infrastructure Emission Database, GID; Wang et al., 2019) and surrounding coal-fired power plants (Global Energy Monitor, GEM; and the Global Power Plant Dataset, World Resources Institute, WRI, Byers et al., 2018)TS4 that support the nearby industries.

Our goal is to calculate $ER_{CO}$ from every satellite sounding within an urban plume, which is a downwind area affected by urban emissions (Sect. 2.2). Sounding-dependent $ER_{CO}$ values are calculated as ratios of CO emissions over $CO_2$ emissions (Eq. 3) that are estimated from satellite-derived fossil fuel (FF) enhancements and further refined with the "scaling factor" in Eq. (1). This scaling factor accounts for several mismatches between OCO-2/3 and TROPOMI (Sect. 2.1) and is obtained from an atmospheric transport model (Sect. 2.2.1). Since we do not differentiate emission signals due to biofuel and fossil fuel combustion, the term "FF enhancement" is used to refer to the *column enhancement induced by any anthropogenic combustion processes in the target city*. The determination of FF enhancements requires estimates of the background values (Sect. 2.2.2) and "second-order" correction terms for biogenic and pyrogenic sources (Sect. 2.2.3). Sounding-specific ERs and uncertainties (Sect. 2.2.4) are aggregated to yield an ER per overpass and per city. Lastly, we illustrate how the $ER_{CO}$ values associated with heavy industry in Los Angeles and Shanghai can be extracted with the assistance of WU-DAPT (Sect. 2.3).

## 2.1   Satellite observations and data pre-processing

We evaluate all coincident OCO-3 SAM and TROPOMI overpass observations, but only select those with relatively small differences in overpass times. Considering the limited number of coincidences between sensors, two non-

SAM overpasses from OCO-3 and one OCO-2 overpass are added to the analysis. As a result, six OCO–TROPOMI co-incidences with high data quality from October 2019 to June 2021 are integrated into the final result for every city. Two of the total of 24 overpasses fall within the Northern Hemisphere summer months (both in June).

### 2.1.1   OCO-2/3 $XCO_2$

The column-averaged dry-air mole fraction of $CO_2$ ($XCO_2$) is retrieved from the reflected sunlight over two $CO_2$ bands centered on 1.6 and 2.0 μm and the oxygen A band for obtaining the surface pressure (Eldering et al., 2019; Taylor et al., 2020). In addition to the standard nadir, glint, and target modes, OCO-3 collects several adjacent swaths of $XCO_2$ observations over a spatial area of approximately 80 km by 80 km during its SAM mode, e.g., four individual swaths in an overpass over LA on 24 February 2020 (Fig. 2a). Similar to OCO-2, each satellite swath comprises eight spatial footprints/soundings, and each sounding has an area of $\sim 1.6 \times 2.2$ km$^2$ at nadir (Fig. 2a). Our analysis only uses screened OCO-3 B10r/B10p4r data (Eldering, 2021) with an $XCO_2$ quality flag of zero (QF = 0). It is worth highlighting that the B10r/B10p4r product is superior to the Early version of OCO-3 (Taylor et al., 2020); it has improved geo-location, advanced radiometric calibration, improved quality filters, and customized post-processing bias correction. As OCO-3 is mounted on the International Space Station, which is in a precessing orbit, its overpass time varies (for example, from 07:00 to 15:00 LT (local time)) for the overpasses we examine, unlike OCO-2.

### 2.1.2   TROPOMI XCO

The TROPOMI column density of CO molecules [mole cm$^{-2}$] is retrieved via the measured radiation from shortwave infrared wavebands centered at $\sim 2.3$ μm (Veefkind et al., 2012). We select soundings with a quality assurance of $\geq 0.5$ as recommended by the TROPOMI README document (Landgraf et al., 2020) and convert the vertical column density to the total column-averaged dry-air mole fraction of CO [XCO in ppb] by calculating the dry-air column density [mole cm$^{-2}$] using the retrieved surface pressure and total column water vapor. TROPOMI CO is retrieved from a larger pixel area of $\sim 7 \times 7$ km$^2$ at nadir, which reduces to $5.5 \times 7$ km$^2$ after 6 October 2019 (Fig. 2c). The overpass time of TROPOMI is $\sim 13:30$ LT for an equatorial overpass in nadir measurements, with a time span of 1–2 h for soundings on the edge of the wide swath (i.e., $\sim 2600$ km).

### 2.1.3   Differences between the two sensors/species

Four mismatches between OCO-3 $XCO_2$ and TROPOMI XCO that pose challenges when extracting FF enhancements

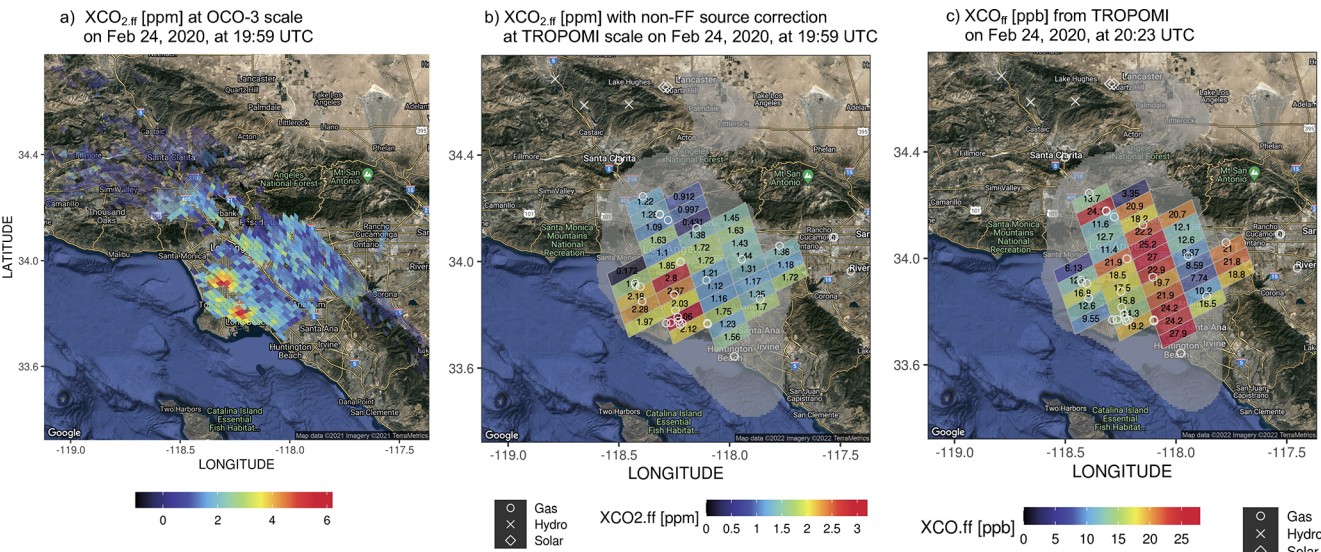

**Figure 2.** Spatial maps of FF $XCO_2$ enhancements with corrections for non-FF sources at the native OCO-3 scale (**a**, ppm) and aggregated at the TROPOMI scale (**b**, ppm) along with FF XCO enhancements (**c**, ppb) over Los Angeles on 24 February 2020. Power stations with different primary fuel types are displayed as different white symbols based on the Global Power Plant Dataset (Byers et al., 2018) TS5. The overall X-STILT column footprint [ppm $(\mu mol \, m^{-2} \, s^{-1})^{-1}$] from all soundings is drawn in light gray (see explanations for the column footprint in Sect. 2.2.1). The underlying hybrid maps were created using the ggmap library in R with the hybrid view from Google Maps over LA (copyright: map data © 2021, imagery © 2021 TerraMetrics).

and $ER_{CO}$ from atmospheric observations are accounted for in this analysis:

1. *Satellite pixel area.* $XCO_2$ enhancements from multiple OCO-2/3 soundings falling within a given TROPOMI polygon are grouped and averaged (Fig. 2a vs. b). For simplicity, the centered lat/long coordinate of an OCO pixel is used to determine its corresponding TROPOMI polygon. The retrieval uncertainty tied to each OCO sounding is also aggregated according to the TROPOMI sampling, contributing to the total observational uncertainty (Sect. 2.2.4).

2. *Averaging kernel profile.* Within the planetary boundary layer, where most emissions occur, TROPOMI XCO retrieval is affected by cloud height/fractions, which yields a lower-than-unity AK (Supplement Fig. S1). The OCO-2/3 $XCO_2$ retrieved under cloudy conditions is typically omitted from Lite files and when QF = 0 is applied; thus, its AK normally approaches 1 near the surface for cloud-free scenes. The mismatch in AK between sensors must be accounted for as it can affect the interpretation of ERs. In this work, we account for AKs within an atmospheric transport model (Sect. 2.2.1).

3. *Overpass times, meteorological conditions, and emission variations.* As a result of the overpass time difference between sensors, variations in meteorological conditions (e.g., wind direction and speed) can lead to changes in the urban plume shapes detected by the two sensors as they pass by. We deal with changes in wind speed and wind direction separately. The former is resolved by using the "scaling factor" inferred from an atmospheric transport model and the latter undergoes manual evaluations (Sect. 3.1). Also, the CO and $CO_2$ emissions themselves can vary over the course of a day, driven by, e.g., the road transportation and residential sectors. Given the overpass time difference between sensors, it is likely that such a mismatch in the timing of CO versus $CO_2$ emissions may affect the observed $ER_{CO}$.

4. *Non-fossil-fuel sources/sinks.* Not accounting for the influences from the biosphere and biomass burning may bias $ER_{CO}$. Given our definition of the "local background", the contrast in non-FF concentration anomalies between the urban and the background regions needs to be included (for more explanation, see Sect. 2.2.3).

## 2.2 Estimates of $E_{gas}$, $ER_{CO}$, and uncertainties

Previous studies (Mitchell et al., 2018; Wu et al., 2020; Lin et al., 2021) proposed an approach for calculating an overall $CO_2$ or $CH_4$ flux using atmospheric measurements and an atmospheric transport model without relying on prior information from emission inventories. Here we briefly describe this approach to obtaining the overall emission of either $CO_2$ (Eq. 1) or CO (Eq. 2) for a single sounding $S$, modified from

Wu et al. (2020):

$$\langle E_{CO_2,s} \rangle = \frac{X_{ffCO_2,s}}{\langle XF_{CO_2,s} \rangle}$$
$$= \frac{X_{obsCO_2,s} - X_{bgCO_2} - \delta X_{bioCO_2,s} - \delta X_{bbCO_2,s}}{\iint XF_{CO_2,s}(x,y)\,dx\,dy}, \quad (1)$$

$$\langle E_{CO,s} \rangle = \frac{X_{ffCO,s}}{\langle XF_{CO,s} \rangle} = \frac{X_{obsCO,s} - X_{bgCO} - \delta X_{bbCO,s}}{\iint XF_{CO,s}(x,y)\,dx\,dy}. \quad (2)$$

All the $X$ terms in the numerator contribute to the estimate of the FF column enhancement ($X_{ff}$, Fig. 2). $X_{ff}$ at a downwind satellite sounding $S$ is the net result of FF sources over the source region $(x,y)$. To describe the source region and attribute it to each satellite sounding, we adopt the column version of the Stochastic Time-Inverted Lagrangian Transport (X-STILT) model (Lin et al., 2003; Fasoli et al., 2018; Wu et al., 2018). This model helps to provide a "scaling factor" $\langle XF_{gas} \rangle$ that accounts for the sounding-specific AK profile and meteorology (Sect. 2.2.1). $X_{bg}$ denotes the local background values from satellite observations uncontaminated by the emission from the city, which are constant for a group of background observations. The background region is usually chosen to be a rural region outside the urban plume while considering the wind direction (Sect. 2.2.2). From a Lagrangian viewpoint, the air parcels arriving at an urban sounding might be traced back to different origins from the air parcels arriving at a rural sounding, meaning that observations at the two soundings may be influenced differently by the surrounding biosphere. Hence, two background correction $\delta$ terms are attached to account for the urban–background gradient in concentration anomalies due to net ecosystem exchange (NEE) and biomass burning (Sect. 2.2.3).

For a given sounding, the estimated flux $\langle E_{gas} \rangle$, with units of $\mu mol\,m^{-2}\,s^{-1}$, represents the average emission over the corresponding source region of that sounding, which should not be confused with the direct emission at that sounding location. The $ER_{CO}$ for a given sounding $S$ is then derived from Eqs. (1) and (2) as follows:

$$ER_{CO,s} = \frac{\langle E_{CO,s} \rangle}{\langle E_{CO_2,s} \rangle} = \frac{X_{ffCO,s}}{X_{ffCO_2,s}} \frac{\langle XF_{CO_2,s} \rangle}{\langle XF_{CO,s} \rangle} = \frac{X_{ffCO,s}}{X_{ffCO_2,s}} \gamma_{foot,s}, \quad (3)$$

where $\frac{X_{ffCO,s}}{X_{ffCO_2,s}}$ is the observed enhancement ratio and $\gamma_{foot,s}$ measures how enhancement ratios with no consideration of AKs and meteorology differ from emission ratios. We simply use ppb-CO/ppm-CO$_2$ TS6 as the units of $ER_{CO}$ (i.e., the same as mmol-CO/mol-CO$_2$ TS7).

## 2.2.1 The X-STILT model

The X-STILT model is adopted in this study (1) to provide the scaling factor $\langle XF_{gas} \rangle$ that resolves differences in AKs and changes in wind speeds, (2) to identify an overpass-specific urban plume for determining background regions (Sect. 2.2.2), and (3) to estimate the sounding-specific biogenic and pyrogenic anomalies for background corrections (Sect. 2.2.3).

STILT releases an ensemble of air parcels from target observations (known as the "receptor") and tracks the movement of those air parcels backward in time. The source region corresponding to each sounding is inferred from the "source–receptor relation" or the STILT "footprint" (Lin et al., 2003; Fasoli et al., 2018). The STILT footprint [ppm ($\mu mol\,m^{-2}\,s^{-1}$)$^{-1}$ TS8] describes the change in atmospheric concentration [ppm] at a downwind location due to possible upwind sources/sinks [$\mu mol\,m^{-2}\,s^{-1}$]. The magnitude of the STILT footprint tends to be higher close to the target observation or under steadier wind conditions; thus, air parcels within the boundary layer can interact more closely with fluxes from the surface.

To accommodate the use of satellite-based column data, X-STILT incorporates retrieval-specific AK and pressure weighting profiles into the footprint calculation (Wu et al., 2018) such that influences on air parcels originating from various altitudes of an atmospheric column are weighted by the sensor/species/sounding-specific vertical profile (Fig. S2). The "column footprint" ($XF_{gas}$) measures the sensitivity of the total column concentration to upwind fluxes from the perspective of a specific satellite sensor. For instance, $XF_{gas}$ for TROPOMI XCO differs from $XF_{gas}$ for OCO-2/3 XCO$_2$, even for concurrent observations. Since the airflow arriving at each satellite observation is unique, the magnitude and spatial distribution of $XF_{gas}$ vary across soundings (Fig. S3). By taking an average of these sounding-dependent column footprints, as shown in Fig. S3, we can identify the source region for all soundings in a SAM (light gray area in Fig. 2b, c). In this work, we only traced air parcels back for 12 h to calculate column footprints, which is sufficient to capture the near-field influence from the target city and better aligned with the local background region outside the city (Sect. 2.2.2).

In short, the spatial summation of column footprints $\langle XF_{gas} \rangle$ is regarded as a scaling factor to address the sounding-specific meteorological conditions and AK profile. The term $\gamma_{foot}$ derived from Eq. (3) reveals the difference between a simple enhancement ratio and a more robust, model-corrected emission ratio.

## 2.2.2 Background definition

Defining accurate background levels to extract urban FF enhancements has always been a challenge in top-down analyses, especially when dealing with column measurements with small signal-to-noise ratios. Wu et al. (2018) compared several approaches to determine a localized XCO$_2$ background for extracting urban signals from OCO-2, including approaches that (1) solely use satellite observations with statistics (e.g., daily median); (2) solely use an atmospheric transport model (e.g., the "curtain method" based on

global concentration fields); and (3) combine observations and transport information from models. Here, we expand the third approach to arrive at localized swath-dependent background values. The broader spatial coverage compared to the narrow swath of OCO-2 and multiple swaths stretching out of the city domain of OCO-3 SAMs help improve such background determination by introducing spatial variations in the background. Accurately describing latitudinal or spatial gradients in the background $XCO_2$ has been emphasized recently (Ye et al., 2020; Schuh et al., 2021).

The process of background determination used in this work involves the first step of identifying the urban plume and differentiating soundings as being within or outside of the plume. To outline the urban plume shape at the overpass time, we utilize the forward mode of STILT with the inclusion of wind uncertainty in atmospheric dispersion. Specifically, 1000 air parcels are released continuously from a rectangle representing the city domain (dashed black box in Fig. 3) every 30 min starting 10 h ahead of the overpass time. All air parcels are allowed to travel forward in time for 12 h from their initial release times. A random wind component typifying model-data wind errors is added to the parcel dispersion (Lin and Gerbig, 2005). We subset the air parcels only during the overpass time and apply a two-dimensional kernel density estimate (KDE) based on the parcels' spatial distributions (blue to purple contours in Fig. 3). KDE is carried out using the kde2d function provided by the MASS library in R (Venables and Ripley, 2002). These normalized KDE contours indicate the likelihood and shape of an urban plume when the satellite scans through. The extent of the urban plume is finalized using a normalized KDE contour of 0.15 (black curve in Fig. 3), which is appropriate to include soundings with a possible influence from the target city and to exclude observations elevated by another city (e.g., the red polygons centered at ∼ 32° N and 120° E in Fig. 3c). This procedure is carried out separately for OCO-2/3 and TROPOMI to reveal the impact of changing meteorology on urban plumes at different overpass times (see Sect. 3.1). It is worth stressing that only enhancements within the urban plume are used for $ER_{CO}$ estimates.

Next, the background value is calculated as the median value of observed $X_{gas}$ per swath over the background region. For example, the background region is the area to the east outside the urban plume since southeasterly wind dominates (Fig. 3b, c). Background values vary with swaths if an OCO-3 SAM is examined. We choose the median instead of the mean to minimize the impact of any "outliers" that may be from a second FF source (other than our target cities) in the background region. Background uncertainty is estimated as a component of the total observed uncertainty (Sect. 2.2.4).

### 2.2.3 Background correction terms for non-FF sources/sinks

The swath-dependent local background approach described above explicitly assumes equal contributions from non-FF sources and sinks for soundings in the background versus soundings in the urban plume, which may not always be the case. We then correct for the spatial gradient in contributions from biogenic and pyrogenic fluxes.

As proposed in Wu et al. (2021), rather than absolute biogenic concentration anomalies, it is the contrast in these anomalies between the background versus the urban plume that is required, considering our localized background definition. Specifically, hourly X-STILT column footprints are convolved, respectively, with hourly mean NEE from a biospheric model representation and daily mean wildfire emissions from the Global Fire Assimilation System (GFAS, Kaiser et al., 2012) to estimate the sounding-specific absolute column anomalies $X_{bio}$ and $X_{bb}$. The Solar-Induced Fluorescence (SIF) for Modeling Urban biogenic Fluxes (SMUrF, Wu et al., 2021) model estimates gross primary production (GPP) from a contiguous SIF product (CSIF, trained based on OCO-2 SIF, Zhang et al., 2018) and respiration based on modeled SIF-based GPP and air and soil temperatures.

Next, the urban–background gradient in such anomalies is calculated as the difference between sounding-specific anomalies and the mean anomaly within the background region:

$$\delta X_{bioCO_2}(s) = X_{bioCO_2}(s) - \overline{X_{bioCO_2}(s_{bg})}, \tag{4}$$

where $s$ or $s_{bg}$ represents all the soundings or select soundings in the background region, respectively. Let us imagine a summer day at noon in the Northern Hemisphere. The urban core is normally associated with a weaker biospheric uptake than the surrounding rural region. Biogenic signals $X_{bioCO_2}(s)$ for soundings in the city are less negative than the mean biogenic signal over the rural background $\overline{X_{bioCO_2}(s_{bg})}$. Hence, the urban–background biogenic gradient $\delta X_{bioCO_2}(s)$ is normally positive and subtracted from the total column (Eq. 1). The estimated $X_{bio}$ values and their urban–background gradient $\delta X_{bio}$ are shown in Sect. 3.1.

Flux exchanges from the ocean and chemical transformations (e.g., the CO sink from the hydroxyl radical (OH) and the source from the oxidation of volatile organic compounds, VOCs) are not considered. The average lifetime of CO against OH ranges from a few weeks to several months depending on the season – much longer than the few-hours timescale we care about. Yet, CO can be generated from the oxidation of $CH_4$ and non-methane VOCs at various rates, which is discussed in Sect. 4.3.

### 2.2.4 Uncertainty sources

The uncertainty related to emissions should contain uncertainties from (1) the atmospheric transport (i.e., column foot-

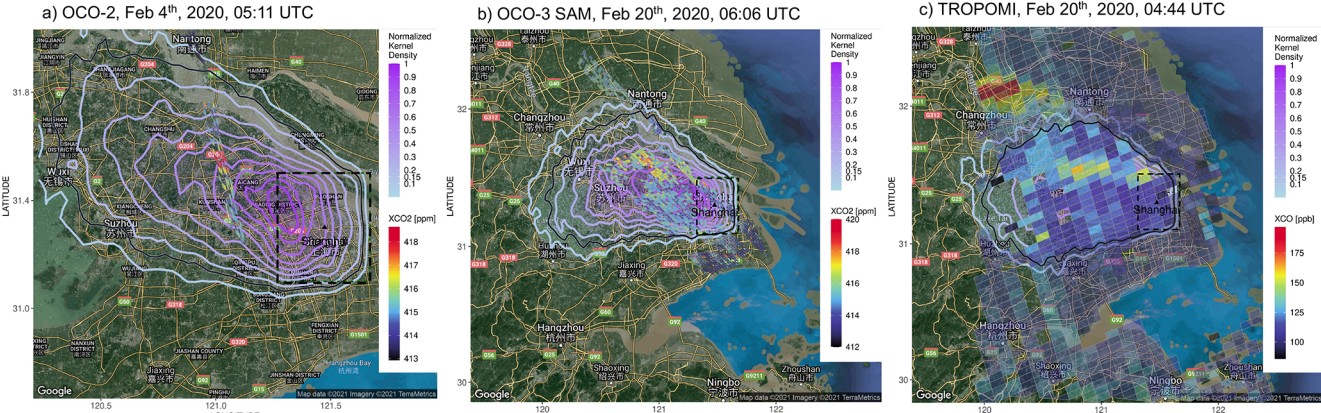

**Figure 3.** Demonstrations of background determination from OCO-2 $XCO_2$ on 4 February 2020 **(a)** and OCO-3 SAM $XCO_2$ [ppm] and TROPOMI XCO [ppb] on 20 February 2020 **(b, c)** over Shanghai. The model-based urban plume (solid black curve) is determined by the normalized 2-D kernel density of the air parcel distribution during the overpass time (blue-purple contours). The "background soundings" outside the urban plume are highlighted with black outlines, while other soundings are outlined in white. For example, OCO-2 observations to the north outside the plume ($\sim 121.1°$ E, $31.9°$ N, **a**) and OCO-3 and TROPOMI soundings to the southeast outside the plume **(b, c)** are used to estimate background values and background uncertainties. The underlying hybrid maps were created using the ggmap library in R with the hybrid view from Google Maps over Shanghai (copyright: map data © 2021, imagery © 2021 TerraMetrics).

prints), (2) observations, and (3) non-FF sources and sinks, according to Eqs. (1) or (2). We neglect uncertainties from column footprints, assuming that no transport bias exists during either the OCO or the TROPOMI overpass time. The urban–background gradient in non-FF fluxes remains very small compared to FF enhancements (Sect. 3.1).

We estimate the uncertainties of observed FF enhancements following Eq. (5). As previously described, observations from a few screened OCO soundings ($\sim 5$ to 28 OCO soundings, depending on the TROPOMI footprint size) are averaged to arrive at a mean $XCO_2$ at the TROPOMI scale. Due to this averaging/binning process, the $XCO_2$ uncertainty due to binning is considered using the standard deviation of $XCO_2$ observations ($\sigma^2_{\varepsilon,\text{bin}}$ in Eq. 5) within a TROPOMI polygon.

$$\sigma^2_{\varepsilon,\text{obs}} = \sigma^2_{\varepsilon,\text{bin}} + \sigma^2_{\varepsilon,\text{bg}} + \sigma^2_{\varepsilon,\text{retrv}}. \qquad (5)$$

Here, $\sigma^2_{\varepsilon,\text{bin}}$ is not required for estimating the XCO uncertainty. The background uncertainty ($\sigma^2_{\varepsilon,\text{bg}}$) contains both the retrieval error and the variability of column observations (as standard deviations) within background regions.

The retrieval uncertainty ($\sigma^2_{\varepsilon,\text{retrv}}$) of XCO is available for each TROPOMI sounding, whereas that of $XCO_2$ is reported for individual OCO-2/3 soundings (as read from Level 2 Lite files), which need to be aggregated at the TROPOMI scale. Due to possible correlations in retrieval errors between nearby OCO soundings, we estimate the error correlation length scale ($L_x$) using exponential variograms, as demonstrated in Fig. S4. Within a TROPOMI polygon that contains $N$ (i.e., the number of) OCO soundings, an error variance–covariance matrix with dimensions of $N \times N$ is constructed with diagonal elements filled with OCO sounding-specific

retrieval error variances. Then, $L_x$ is used to form the normalized covariance matrix, i.e., $\exp(-\frac{D(S_i, S_j)}{L_x})$, where $D(S_i, S_j)$ denotes the distance between each two OCO soundings ($1 \le i < j \le N$). Lastly, the sum of all elements in the error covariance matrix (both variance and covariance elements) is divided by $N^2$ to obtain one $\sigma^2_{\varepsilon,\text{retrv}}$ per TROPOMI grid. As a result, the overall uncertainty of FF enhancement per sounding is often dominated by the background error component.

## 2.3 Identifying the $ER_{CO}$ for heavy industry within a city

A key objective of this study is to explain the intra-city variability of $ER_{CO}$ by exploring sector-specific or sector-dominant combustion activities. While certain combustion processes and sectors tend to have higher ERs than others, sectorally dependent ERs are variable within and across cities. The ERs derived from atmospheric observations comprise a mixed effect of different activities in the city. Previous attempts include reducing the number of sectors and relying on prior sector-specific ERs via a (joint) Bayesian inversion (Brioude et al., 2012; Nathan et al., 2018).

Here, we propose a novel approach to identifying ERs associated with heavy industry in a city. Instead of relying on prior emission inventories that can sometimes be erroneous regarding the magnitudes and the locations of sector-specific activities (see discussions in Sect. 4.4), we utilized an urban land cover classification dataset, WUDAPT, that provides Local Climate Zone (LCZ) classifications at a grid spacing of 120 m (Ching et al., 2018). As shown in Fig. 4a, d, LCZ categories include street canyons (e.g., compact/open/lightweight, high/mid/low rise), building spacing

(e.g., sparsely built, heavy industry), and tree spacing (e.g., dense/scattered trees, low plants, rocks, etc.). Each LCZ is unique in its thermal, radiative, and metabolic properties. For instance, the compact high-rise (LCZ 1) and heavy industry (LCZ 10) categories have the highest anthropogenic heat outputs of 50–300 and $> 300\,\mathrm{W\,m^{-2}}$, respectively (Stewart and Oke, 2012). Heavy industry is defined as low-rise and midrise industrial structures (towers, tanks, stacks) and mostly paved or hard-packed metal with steel and concrete construction materials and few or no trees in WUDAPT (Ching et al., 2018), which differs from the industry-relevant sectors defined by the Intergovernmental Panel on Climate Change (e.g., as used in EDGAR). We clarify that we are not trying to tackle individual industrial processes, which is much more difficult. As of this analysis, LCZ maps are only available for a limited number of cities, including Shanghai and LA, but they have recently been generalized to the entire globe (Demuzere et al., 2022a).

To relate $ER_{CO}$ to heavy industry, the percentage of heavy industry is first interpolated using 1 km grid spacing from WUDAPT LZC maps (%, Fig. 4c, f). The industrial coverage map is then convolved with the X-STILT column footprint (Fig. S3) to quantify the industrial influence on each TROPOMI polygon $P_{ind}(x, y)$, which is defined as the column footprint-normalized industry fraction (Fig. S5). For example, soundings in the city center farther away from the heavy industry in LA are related to smaller influences. Lastly, we sum $P_{ind}(x, y)$ across the space to arrive at $\langle P_{ind} \rangle$, which serves as a metric of how much the observation at a given sounding is affected by heavy industry. Specifically, soundings with $\langle P_{ind} \rangle$ larger than the 75th or 90th percentile are marked as locations that are "impacted" or "strongly impacted" by heavy industry within the city. Sensitivity and significance analyses are conducted and presented in Sect. 3.2.2; these test if: (1) the results are subject to the percentile threshold when defining industry-dominated soundings; and (2) ERs over industry-dominated soundings are statistically significantly different from ERs for the remaining soundings.

## 3 Results

$ER_{CO}$ values and uncertainties are reported at multiple spatial scales, from the spatially resolved sounding level (Eq. 3) to the overall overpass and city level. Again, only ERs at soundings within the urban plume are selected. Overpasses with too few valid soundings in a plume area are also removed from the results. Before presenting ERs at different spatial scales, we assess factors that may influence the derived $ER_{CO}$.

### 3.1 Interference factors that modify $ER_{CO}$

We examine impacts on $ER_{CO}$ from the following interference factors: (a) differences in AKs between OCO-2/3 $XCO_2$ and TROPOMI XCO; (b) shifts in wind fields between two overpass times; (c) the urban–background contrast in biogenic and pyrogenic contributions; and (d) temporal variation in the emissions themselves. In summary, we find that differences in AKs and wind directions between sensors can significantly affect the spatially resolved $ER_{CO}$. For the final 24 overpasses we selected, temporal variations in the emission pattern and urban–background gradients in biogenic/pyrogenic contributions play minor roles in overpass- or city-level ERs.

Recall that sounding-specific AKs and wind speeds were considered in the sounding-specific column footprint using X-STILT (Sect. 2.2), and $\gamma_{foot} = \frac{\langle XF_{CO_2} \rangle}{\langle XF_{CO} \rangle}$ measures the overall contributions from AKs and wind speeds to the spatially resolved $ER_{CO}$ (Supplement Fig. S6c). For instance, the mean $\gamma_{foot}$ spans from 1.20 to 1.57 over LA and from 1.02 to 1.38 over Shanghai across different overpasses (printed in Fig. 7a, c). $\gamma_{foot}$ is generally larger than 1 because AKs of TROPOMI XCO near the surface are smaller than surface AKs for OCO-2/3. Simply using enhancement ratios without accounting for mismatches in AKs and wind speeds between sensors will likely lead to an underestimation of emission ratios (Eq. 3). On average, the overpass-level $ER_{CO}$ can be $\sim 20\,\%$ higher than enhancement ratios across our 24 overpasses.

The second factor is the change in wind directions between two overpass times, which is evaluated using the same algorithm as the urban plume detection in Sect. 2.2.2. Again, colored contours and curves in Fig. 5 indicate neither the intensities of concentrations nor flux fields (as no prior emissions are used) but rather the likelihood of urban plumes, determined by atmospheric dispersion with random wind uncertainties. Matching between OCO-3 soundings and TROPOMI polygons as described earlier would be fine for concurrent observations (Fig. 5a, b), but this becomes problematic if $\Delta t$ becomes large ("outliers" with significant changes in plumes in Fig. 5e, f). The cases between the good cases and the outliers are to be used with caution (Fig. 5c, d). By comparing the overlap of plumes at the two times, we shifted OCO-3 soundings to better align with TROPOMI polygons. For example, on 20 February 2020, because the modeled plume at the OCO-3 overpass time (06:06 UTC) appears northward compared to the plume at the TROPOMI overpass time (04:44 UTC), OCO-3 soundings were shifted southward by zero to two grids, depending on their longitudinal coordinates (Fig. S7). In other words, by shifting the FF $XCO_2$ enhancements, we better align the urban plume at the OCO-3 time with the plume at the TROPOMI time. Every OCO–TROPOMI coincidence is manually examined and assigned to one of the three categories, which are further summarized in Sect. 4.1. Outliers are removed from this analysis, since no simple wind or plume rotation would improve their $ER_{CO}$ estimates.

Besides changes in wind directions, the CO and $CO_2$ emissions themselves can vary across daytime hours, likely driven by the road transportation and residential sectors. As a result,

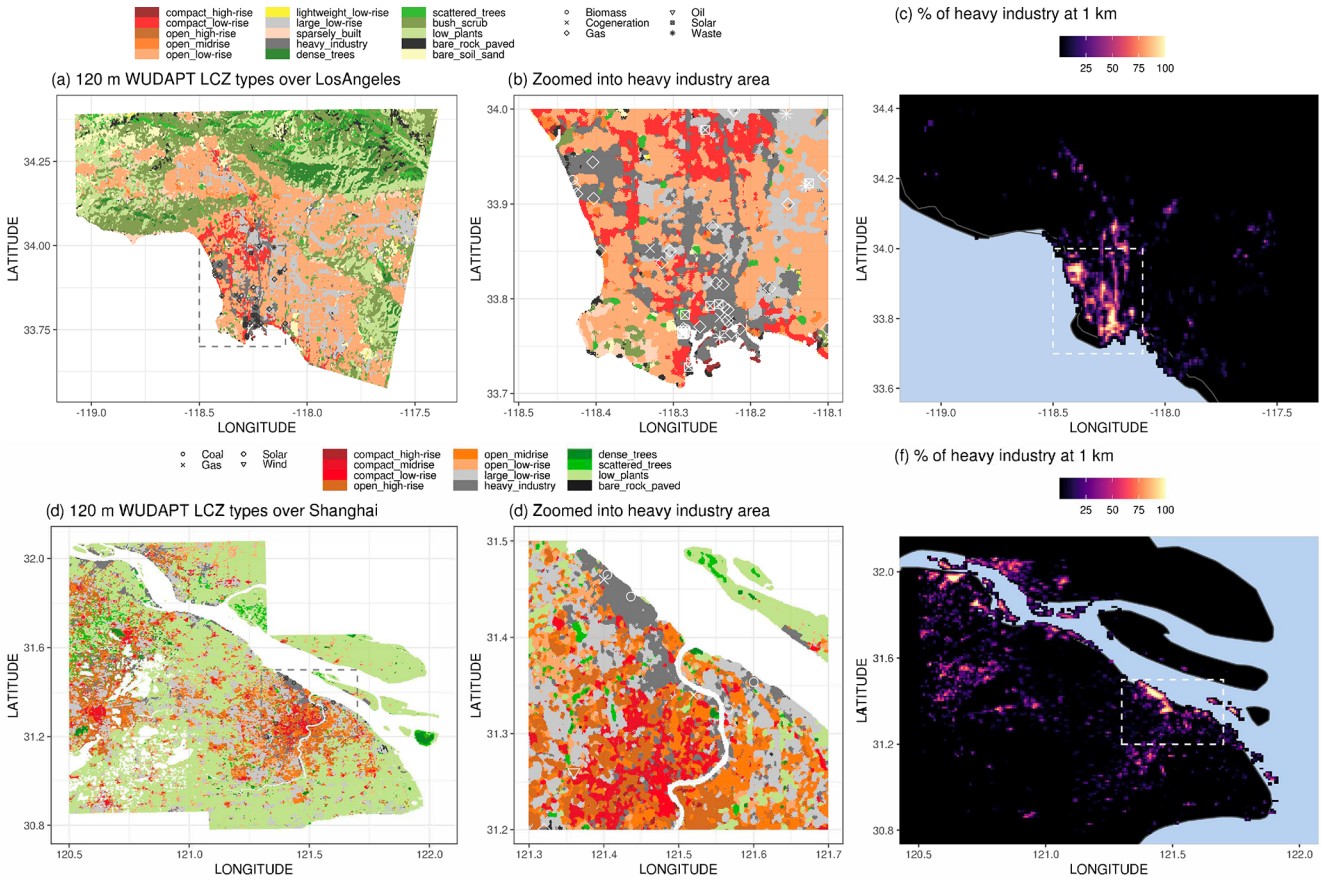

**Figure 4.** Maps of 120 m Local Climate Zone (LCZ) from WUDAPT (**a** and **d**) along with magnified images (**b** and **e**) and interpolated areal coverage of the heavy industry [%] at 1 km around Shanghai and Los Angeles (**c, f**). LCZ classifications centered on Wuxi and Shanghai are combined. Based on the Global Power Plant Dataset (Byers et al., 2018) TS9, power stations are drawn as white symbols. The dashed light gray or white rectangles in the maps indicate the magnified region.

variations in the derived $ER_{CO}$ across multiple overpasses may reflect not only the variation in combustion efficiencies but also the mismatch in the emission timing. LA may be a city with more distinct daytime changes in emissions compared to industry-centered cities. Fortunately, based on a supplementary sensitivity analysis using measurements from the Total Carbon Column Observing Network in Pasadena (TC-CON, Wennberg et al., 2017), observed $ER_{CO}$ values appear to be less variable when limiting satellite overpasses to those with a smaller time difference (Fig. S8). Future geostationary satellite monitoring of $NO_x$ (e.g., TEMPO, Chance et al., 2022) may provide better guidance regarding the hourly pattern in urban emissions, especially from the traffic sector, which show more daytime fluctuations, as discovered using surface monitoring networks (e.g., over Chicago; de Foy, 2018).

The last factor is the urban–background contrast in contributions from non-FF sources and sinks. The biogenic $XCO_2$ anomaly modeled using SMUrF and X-STILT ranges from −0.7 to 0.3 ppm per OCO-3 sounding, depending on the hour of the day (i.e., the solar zenith angle), season, and wind direction (Fig. S9). As explained in Sect. 2.2.2, urban–background gradients in these biogenic anomalies (i.e., $\delta X_{bio}$) were used to correct the constant localized background $X_{bg}$ (Eq. 1). Take the two overpasses with the largest urban–background contrast as examples: as biospheric uptake is normally weaker in urban areas than in surrounding rural areas (often used as background regions), the urban–rural gradient for locations in the plume region becomes more positive (Fig. S10b). Nonetheless, even for the one summertime SAM over Zibo on 21 June 2020, the sounding-level $\delta X_{bio}$ ranges from 0 to 0.4 ppm, which remains small compared to the sounding-level $FFCO_2$ enhancements of 2 to 7 ppm (Fig. S11a, b). For most other overpasses, $\delta X_{bio}$ aggregated according to the TROPOMI sampling stays low, with an absolute value of < 0.3 ppm (as printed in each panel of Figs. 6 and 7). Even with a bias in the resultant $\delta X_{bio}$ resulting from an incorrect prior NEE, the effect on the derived FF enhancements and $ER_{CO}$ would be small.

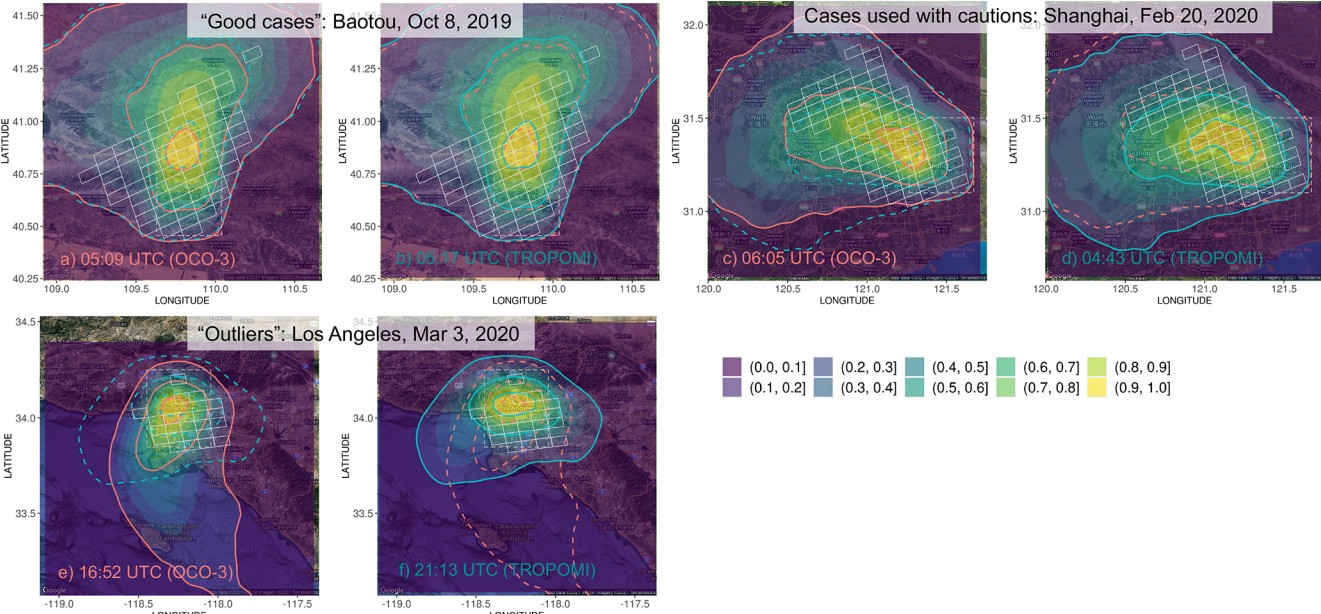

**Figure 5.** Examples of modeled urban plumes during OCO-3 (red curve) and TROPOMI (blue curve) overpass times (in UTC). The likelihood of these meteorology-only urban plumes (no emission involved) is quantified by the normalized KDE binned in 10 intervals of the modeled air parcel distribution (yellow-green-purple contours). Three types of overpasses are shown, as follows: **(a, b)** "good cases" with almost identical urban plumes at two times, e.g., Baotou on 8 October 2019 with a $\Delta t$ of 8 min; **(c, d)** cases to be used with caution, where the urban plume shifts from one time to another and requires a simple plume rotation, e.g., Shanghai on 20 February 2020 with a $\Delta t$ of 1.5 h; **(e, f)** outliers where two urban plumes change significantly, e.g., for Los Angeles on 3 March 2020 with a $\Delta t$ of over 4 h. The underlying hybrid maps were created using the ggmap library in R with the hybrid view from Google Maps over LA (copyright: map data © 2021, imagery © 2021 TerraMetrics).

Although LA is surrounded by occasional intense wildfire activities, the column anomalies due to biomass burning that are suggested by the coupling of GFAS and X-STILT are minimal for the dates we examined. Yet, since wildfire-related $ER_{CO}$ values are usually higher than the FF-related $ER_{CO}$ (Fig. 1), properly accounting for pyrogenic contributions and gradients between urban and surrounding rural areas is important for cities in mountainous and forested areas during fire seasons. For instance, Crounse et al. (2009) leveraged aircraft measurements of HCN and $C_2H_2$ over Mexico City as indicators to disentangle the CO signals due to biomass burning and urban emissions, respectively.

### 3.2 Intra-city $ER_{CO}$ variations and signals from heavy industry

Observed enhancements are the net consequence of associated sources/sinks from source regions. That is, a high atmospheric content of $CO_2$ or CO at the sounding location does not necessarily indicate a high emission rate at this location (Kiel et al., 2021). Our derived emissions and ERs, although reported for each sounding, indicate the overall emission and combustion efficiency over its source region.

In the following subsections, we present ERs for each sounding and the aggregate for each overpass and city. Since the aggregation of sounding-level ERs to a single value per overpass or city is sensitive to the method/statistic adopted, we bootstrapped $E_{CO}$ and $E_{CO_2}$ based on their sounding-specific values and uncertainties to generate a linear regression fit per bootstrap loop (light gray lines in Fig. 6). Specifically, 1000 random sets of $E_{CO}$ and $E_{CO_2}$ were generated according to assumed normal distributions, where sounding-level emission estimates provided mean statistics with observational uncertainties as standard deviations. We used the standardized major axis (SMA) solution for linear regression to minimize deviations of data points from the regression line for both axes. Eventually, we obtained 1000 bootstrapped slopes and selected slopes with positive values, which yielded the overpass-level $ER_{CO}$ and uncertainty (e.g., dashed colored lines and text in Fig. 6). Also, sounding-level $ER_{CO}$ values from all overpasses are presented in histograms and generally follow a log-normal distribution (Fig. 7b, d).

### 3.2.1 Baotou and Zibo

Combustion efficiencies are generally poor for the two industry- and energy-oriented cities. The overpass-specific ERs span from $9.3 \pm 1.2$ to $24.6 \pm 3.8 \, \mathrm{mmol \, mol^{-1}}$, with an integrated city-level estimate of $17.3 \pm 0.5 \, \mathrm{mmol \, mol^{-1}}$ for Baotou (Fig. 6a). According to GID, the Baotou Iron

**Table 1.** A summary of the total power generation capacity (from the Global Power Plant Dataset of the World Resources Institute, Byers et al., 2018 TS10) and information on heavy industry, including the annual crude iron capacity (GID, Wang et al., 2019). Power plants are selected from a $0.5° \times 0.5°$ region around each city; percentages indicate the partitioning by fuel type.

| City | Total power capacity (MW) and partitioning by fuel type | Key industry OR annual crude steel capacity ($kt\,yr^{-1}$) |
| --- | --- | --- |
| Los Angeles | 5808 MW (95.6 % fueled by gas; 0 % by coal) | Refinery, shipping |
| Shanghai | 16 031 MW (75.2 % fueled by coal; 24.4 % by gas) | Iron & steel (25 099 $kt\,yr^{-1}$) |
| Baotou | 6470 MW (100 % fueled by coal) | Iron & steel (12 619 $kt\,yr^{-1}$) |
| Zibo (w/ Zouping) | 9720 MW (100 % fueled by coal) | Electrolytic aluminum; iron & steel (2532 $kt\,yr^{-1}$) |

and Steel Group is located within the city and contributes to an annual capacity of crude iron of $12\,619\,kt\,yr^{-1}$, with estimated $CO_2$ emissions of $20\,462\,kt\,yr^{-1}$ (Table 1). The slightly lower $ER_{CO}$ and FF enhancements in February 2021 coincide with the timing of the Spring Festival in 2021 ($\sim$ 12 February). Standard deviations of bootstrapped slopes are higher for overpasses with fewer high-quality satellite soundings, e.g., $3.8\,mmol\,mol^{-1}$ for overpasses with seven available TROPOMI polygons in the urban plume on 31 May 2020. Utilizing the bootstrap method helps to account for the impact of the sounding number on the overall city-level ER estimate.

Zibo, along with the nearby county-level city of Zouping, accounted for over one-eighth of the total coal consumption of Shandong Province in 2017. The coal-fired power plants in the area contribute to a total power generation capacity of $9720\,MW$ (Table 1), which is likely to support local metal industries, especially the producers of electrolytic aluminum (they are the world's top producers). The maximum $XCO_2$ enhancement per OCO-3 sounding can even reach up to 10 ppm for a few overpasses (not shown). Interestingly, the $ER_{CO}$ for Zibo first declined from $10.1 \pm 1.1$ to $6.1 \pm 0.6\,mmol\,mol^{-1}$ during February 2020 and then gradually increased back to $18.2 \pm 1.1\,mmol\,mol^{-1}$ by June 2020 (Fig. 6b). Such temporal variations in $ER_{CO}$ agree nicely with the timing of the initial phase of the COVID-19 lockdown in China (i.e., February to May 2020) (e.g., Laughner et al., 2021). We suspect that changes in $ER_{CO}$ could be driven by the partial shutdown and reopening of the multiple coal-fired power plants and metal industries in the area.

### 3.2.2 Los Angeles and Shanghai

Although OCO-3 has sampled the Los Angeles Basin dozens of times to date, many overpasses did not pass the quality check (i.e., QF) and were removed from the final result due to their noticeable shifts in urban plumes between two overpass times (e.g., 3 March, 15 April, and 5 May 2020 for LA; discussed in Sect. 4.1). The overpass-level ER ranges from $7.4 \pm 0.8$ to $11.7 \pm 1.5$ mmol-CO $mol$-$CO_2$ TS11, with a city-level value of either $9.6 \pm 0.5\,mmol\,mol^{-1}$ (obtained using the regression approach; Fig. 7a) or $9.7\,mmol\,mol^{-1}$ (obtained using the histogram approach; Fig. 7b). Our space-

based $ER_{CO}$ estimates over LA fall within the range of 7.1 to $12.4\,mmol\,mol^{-1}$ reported from prior studies (Wennberg et al., 2012; Brioude et al., 2013; Hedelius et al., 2016; Silva and Arellano, 2017). Small discrepancies in $ER_{CO}$ between studies may be attributed to discrepancies in the times of interest, sampling strategies, and techniques used for $ER_{CO}$ calculations (e.g., background definition).

In contrast to LA, where urban plumes are usually well constrained by the basin, wind speeds and directions vary across different overpasses over Shanghai – i.e., there is a southeasterly wind on 4 February and 20 February 2020, a southwesterly wind on 24 February 2020 and 19 February 2021, and a northerly wind on 23 April and 30 December 2020. Such changes in the wind regime between overpasses over Shanghai suggest that soundings from an individual overpass may reflect emission patterns over different source regions, which emphasizes the importance of integrating atmospheric transport when interpreting temporal variations in observation-based ERs. In other words, one cannot simply use all the soundings over a city to calculate ERs; it is necessary to select those soundings that are affected by emissions from that city. The overpass-specific ER ranges from $4.2 \pm 1.2$ to $17.1 \pm 6.2\,mmol\,mol^{-1}$, with a city-level average of $10.2 \pm 0.4\,mmol\,mol^{-1}$ based on the linear regression approach (Fig. 7c) or $12.9\,mmol\,mol^{-1}$ using the histogram approach (Fig. 7d).

Now we focus on the distribution of sounding-level ERs for these two megacities (Fig. 7b, d) to see if ERs associated with a part of a city (i.e., the heavy industry region) can be revealed. As described earlier, to locate the soundings affected or strongly affected by the heavy industry in a city while accounting for the overpass-specific meteorology, we coupled the LZC-based industrial coverage (Fig. 4c, f) with X-STILT column footprints and quantified the industrial influence, $\langle P_{ind} \rangle$, at each sounding location.

Industrial regions within the LA Basin are concentrated to the south, near the Port of LA; and to the west of downtown, near Los Angeles Airport and the Chevron Refinery in El Segundo (Fig. 4e). The distribution of ERs for industry-dominated soundings tends to shift slightly towards the lower end (blue or red bars in Fig. 7b) compared to the distribution for all soundings (gray bars in Fig. 7b). For example, ERs of $> 15\,ppb\,ppm^{-1}$ are less frequently found for industry-

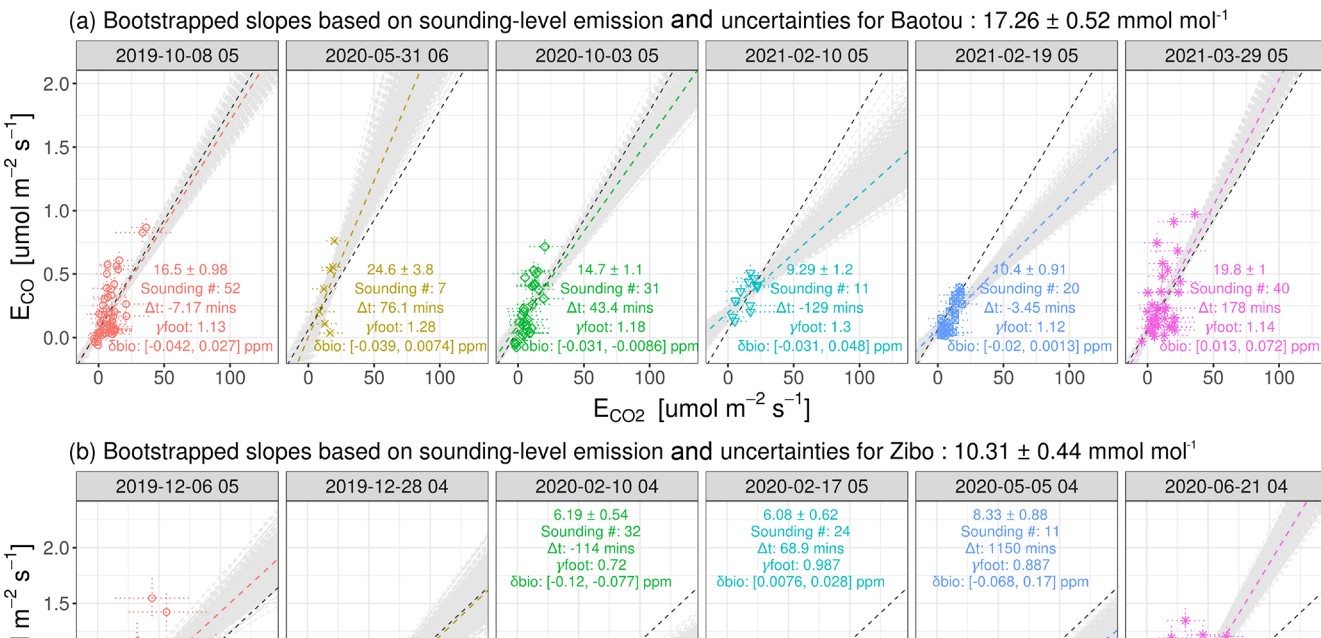

**Figure 6.** Scatter plots of CO and $CO_2$ fluxes [$\mu$mol m$^{-2}$ s$^{-1}$] and their uncertainties (error bars as dotted lines) for Baotou **(a)** and Zibo **(b)**. Linear regressions are applied to the data from each overpass (dashed colored lines) and from all overpasses (dashed black lines). Only bootstrapped regression lines with positive slopes were chosen from the Monte Carlo experiment (dashed gray lines; $\sim 98.4\%$ and 93.3% of the total 6000 bootstrapped lines for Baotou and Zibo, respectively). The TROPOMI overpass time (in UTC), the total TROPOMI sounding number, the discrepancy in overpass times ($\Delta t$, min), the impact of the AK and wind conditions between sensors ($\gamma_{foot}$, unitless), and the range of the urban–rural biogenic gradient ($\delta X_{bio}$, ppm) are labeled in each panel. $XCO_2$ values on 6 December and 28 December 2019 over Zibo came from the non-SAM nadir OCO-3 observations.

dominated soundings (red bars in Fig. 7b). The industry-oriented soundings generally have slightly lower CO but higher $CO_2$ enhancements (Fig. 2b, c) compared to other soundings within the basin, resulting in slightly lower $ER_{CO}$.
No iron and steel facilities or coal-fired power plants are found over the heavy industry area in LA according to GID and GEM. We hypothesize that the slight shift of $ER_{CO}$ towards the lower end may be explained by the heavy-duty diesel engines and natural gas power plants in the Port of LA versus the predominately gasoline vehicles across the city, because the $ER_{CO}$ for heavy-duty diesel vehicles and non-coal-fired power plants is generally lower than that for light-duty gasoline vehicles. For example, a field campaign in 2007 in Beijing that split observations into daytime versus nighttime observations suggested that the ER linked to nighttime diesel transportation is much lower than that for the gasoline sub-sector (Westerdahl et al., 2009, Fig. 1a). Similar to LA, a higher fuel efficiency was found over the ship channel of Houston (ER of $\sim 4$ ppb ppm$^{-1}$) compared to down-

town Houston (ER of $\sim 10$ ppb ppm$^{-1}$) (Brioude et al., 2012, Fig. 1b). Unfortunately, only two good SAMs near Houston from late 2019 to June 2021 are available, but future work can further validate the urban-industry contrast in ERs from space.

In Shanghai, the heavy industry is concentrated to the north of the city center (Fig. 4a). Interestingly, in contrast to LA, ERs affected by heavy industry are skewed towards the higher end, with medians of 16.8 or 18.8 ppb ppm$^{-1}$ (blue or red bars in Fig. 7d) compared to the city-level median of 12.9 ppb ppm$^{-1}$ (black bars in Fig. 7d). CO and $CO_2$ enhancements and $ER_{CO}$ are all higher for industry-oriented soundings than for all soundings combined. Such spatial divergence in enhancements and ERs between heavy industry and the entire city may be attributed to substantial CO emissions from iron and steel production. Schneising et al. (2019) also found that many hotspots with high TROPOMI CO enhancements in China and India are tied to the iron and steel industries. During their production processes, iron ores are

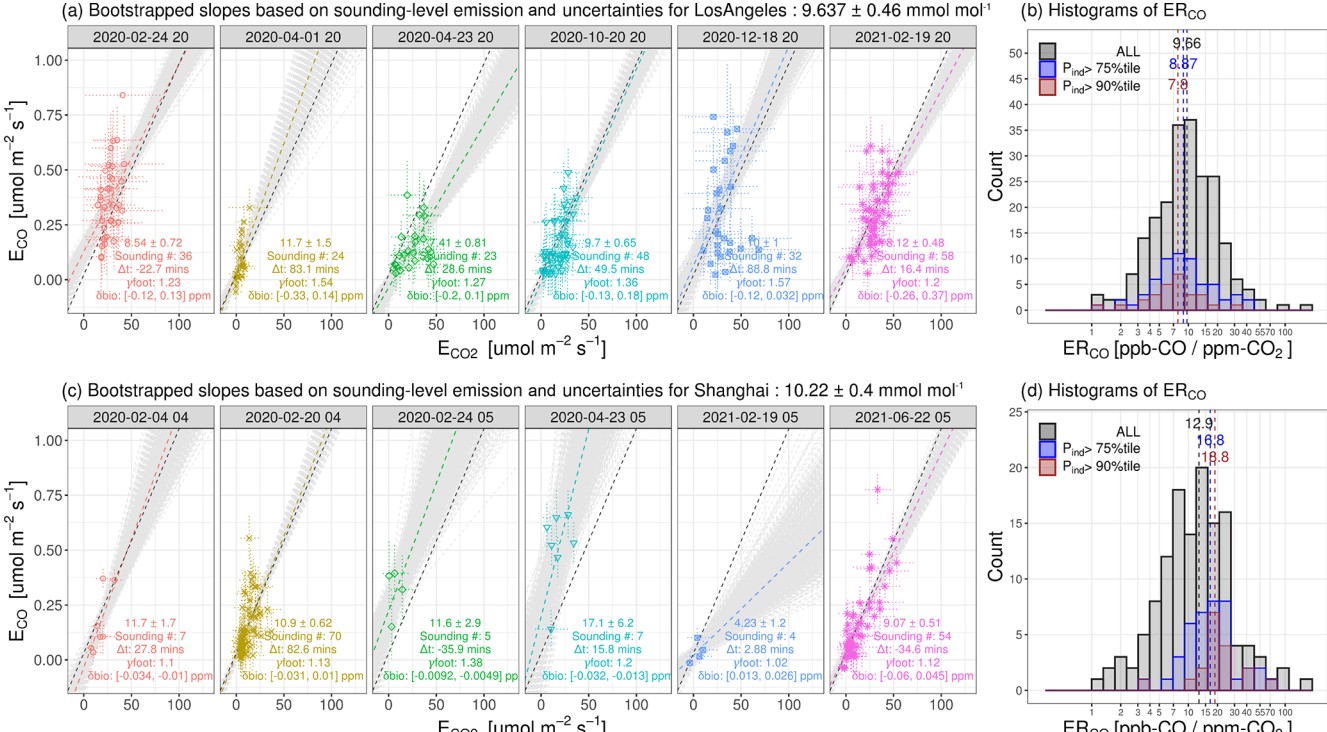

**Figure 7. (a, c)** Same as Fig. 6 but for LA and Shanghai. Only bootstrapped regression lines with positive slopes are presented as light gray lines (∼ 94.3 % and 88.4 % of the 1000 bootstrapped lines per overpass for LA and Shanghai, respectively). **(b, d)** Histogram of $ER_{CO}$ for all soundings (black bars). Soundings impacted or strongly impacted by heavy industry are defined as having $\langle P_{ind} \rangle$ larger than their 75th or 90th percentile (blue or red bars); the corresponding median ERs are also shown (vertical dashed lines). The industrial impact is quantified using column footprints from X-STILT (to account for the atmospheric transport) together with localized information from the urban land cover data WUDAPT.

reduced to crude iron and steel, with CO involved. According to a plant-level estimate in 2019 from GID, Baoshan Iron & Steel Co., Ltd., located to the north of downtown Shanghai, has an annual crude steel capacity of 25 099 kt yr$^{-1}$ (Ta-
ble 1) and a total $CO_2$ emission of 32 148 kt yr$^{-1}$ for all coke, sinter, iron, and crude steel combined.

To validate the robustness of such ER shifts related to heavy industry, we tested the use of different percentile thresholds other than the 75th and 95th percentiles to deter-
mine industry-dominated soundings (Sect. 2.3). The above statements on industry-impacted $ER_{CO}$ hold if using alternative thresholds, including the 50th, 60th, and 80th percentiles. An additional Welch two-sample $t$-test confirms that ERs from industry-dominated soundings significantly differ
from the remaining soundings that are less affected by heavy industry. When the adopted percentile threshold increases from the 50th to the 95th, the divergence in ERs between industrial and non-industrial soundings becomes more apparent, and the $p$ value for the statistical significance of this dif-
ference becomes smaller ($p$ values are < 0.05 for all thresholds). In addition, the average number of OCO-3 soundings in a TROPOMI polygon is roughly the same for industry-

affected soundings versus the rest (e.g., 11.8 vs. 10.3 for LA and 7.3 vs. 8.7 for Shanghai).

We acknowledge that although many iron/steel plants may   25
aim at combusting as much CO as possible before releasing CO into the atmosphere, the indispensable role that CO plays in the iron/steel industry makes it unique when assessing its $ER_{CO}$ and combustion efficiency for various industrial processes. Furthermore, it is difficult to separate combustion   30
signals of individual sectors from observations without prior assumptions of sector-specific contributions, since the atmospheric concentration at a given location arises from various underlying combustion processes spread over the source region. Even using additional co-emitted species, it would be   35
risky to assume that a co-emitted species (e.g., CO or $NO_x$) comes solely from one individual emission sector.

## 4  Discussion

This study is one of the first to analyze intra-city variations of emission ratios between CO and $CO_2$ using two   40
asynchronous satellite sensors. We describe complications induced by discrepancies between satellite sensors and retrievals and demonstrate methods to mitigate these compli-

cations by accounting for sounding-specific averaging kernels, atmospheric transport, and urban–background contrast in the contribution from non-FF sources/sinks using an atmospheric transport model.

## 4.1 Influences from non-FF components and atmospheric transport

Pyrogenic anomalies are minimal for the overpasses we examined but should be considered for certain cities (e.g., during dry seasons over Mexico City, Crounse et al., 2009), considering the high $ER_{CO}$ from forest wildfires of 35 to 80 ppb ppm$^{-1}$ (Fig. 1a). Most overpasses we analyzed fall within the dormant seasons. For the three overpasses during the growing season, biogenic anomalies modeled using the SMUrF model for a given OCO-2/3 sounding may reach up to 0.5 ppm (Fig. S9). Even though modeled NEE and resultant biogenic contributions/gradient can be uncertain, we stress again that it is the urban–background biogenic contrast ($\delta X_{bio}$ in Eq. 1) that is important for estimating FFCO$_2$ enhancements, given our setup for a local background value. Satellite missions such as TROPOMI and the upcoming Geostationary Carbon Cycle Observatory (GeoCarb) will provide solar-induced fluorescence (SIF), which may help improve spatially explicit SIF-based GPP and NEE estimates (Turner et al., 2020; Wu et al., 2021), specifically by reducing the dependence on other remote sensing products and the assumption of model parameters for each plant functional type.

The biggest challenge affecting the robust estimation of spatially resolved $ER_{CO}$ is the shift in wind directions between two overpass times. Substantial changes in wind directions and urban plumes (e.g., the "outliers" in Fig. 5e, f) were mostly found for overpasses with an absolute time difference $|\Delta t|$ of > 2 h (implied by the bars labeled with an asterisk in Fig. 8). If TROPOMI pixel sizes are relatively large (i.e., non-nadir observations) or the wind is steadier, this $|\Delta t|$ constraint may be relaxed as long as emissions for a specific city are less driven by sectors with noticeable diurnal cycles (e.g., road transportation). For instance, on 31 May 2020, TROPOMI polygon sizes for the industry-dominated city Baotou are sufficiently large compared to the shift in urban plumes, despite its $|\Delta t|$ of 3 h (Fig. 8). In addition, we manually re-positioned the OCO-3 soundings to TROPOMI polygons for a few cases (the bars with nonzero numbers on top in Fig. 8) using the simple wind/plume shift demonstrated in Sect. 3.1. Fortunately, future geostationary satellites will be capable of mapping XCO and XCO$_2$ simultaneously at a higher temporal frequency, which will eliminate this issue.

## 4.2 ERs for an individual sounding, overpass, city, and heavy industry region within a city

Contrary to previous work relying on inventory-based sector-based ERs, we attribute the intra-urban gradient to heavy industry using an urban land cover classification dataset. Such high-resolution localized maps help identify the observations strongly influenced by heavy industry. Based on a limited sample size, the heavy industry within the Greater Shanghai area is tied to an $ER_{CO}$ that is higher than the city average, reflecting poorer combustion efficiency (Fig. 7d). Industry- and energy-centered cities such as Baotou and Zibo are less efficient in their combustion activities. In particular, the industry-dominated ER over Shanghai (18.8 mmol mol$^{-1}$, as indicated by the dashed red line in Fig. 7d) aligns better with the overall city-scale ER over Baotou of 17.3 mmol mol$^{-1}$ (Fig. 6). The previously reported urban-integrated $ER_{CO}$ values are mostly constrained within the range of 5 to 20 ppb ppm$^{-1}$, with a few exceptions in East Asia before 2010 that are over 30 ppb ppm$^{-1}$ (Fig. 1). Our city-level estimates from space agree well with the range of previously reported values.

A city-scale $ER_{CO}$ derived from spatially explicit ERs can be influenced by (1) the adopted statistic, (2) overpass dates and overpass-specific wind conditions, and (3) estimated uncertainties in $ER_{CO}$. For example, the overall ER derived from all soundings within the urban plume differs from that derived from a selection of soundings. Even though we started with all quantified OCO-2/3 observations in a SAM, only those located within the urban plumes (black curve in Fig. 3) can be used to estimate ERs, as this allows ERs from overpasses with different meteorological conditions to be compared in an unbiased way. The mean or median value of the sounding-level $ER_{CO}$ (e.g., 13.4 or 9.6 ppb ppm$^{-1}$ for LA in Fig. S12) differs slightly from the city average obtained using the regression slope method when observational uncertainties are taken into account (e.g., 9.6 ppb ppm$^{-1}$ for LA in Fig. 7a). Apart from these bulk quantities, the distribution of $ER_{CO}$ in the linear space is negatively skewed and roughly follows the log-normal distribution (Fig. S12), where a few observations with higher $ER_{CO}$ values are influenced by point sources with poorer combustion efficiency. More observations with finer satellite pixels across the city would improve the robustness of both the spatial distribution and bulk estimates of ERs.

## 4.3 Limitation

The main limitation of this work is the relatively low sample size, which is largely constrained by the requirement for small differences in overpass times. When more satellite data or upcoming data from geostationary satellites become accessible, intra-city ERs can be used to more robustly assess the temporal variation in sector-oriented combustion efficiency, including across seasons or times (e.g., business-as-usual scenarios versus pandemic-disturbed time frames). Beyond the sheer number of soundings, uncertainty arises when aggregating CO$_2$ enhancements from the finer-resolution OCO-3 grid to the TROPOMI sampling. The centered lat/long coordinates of OCO-2/3 soundings are chosen to determine the corresponding TROPOMI polygon, while

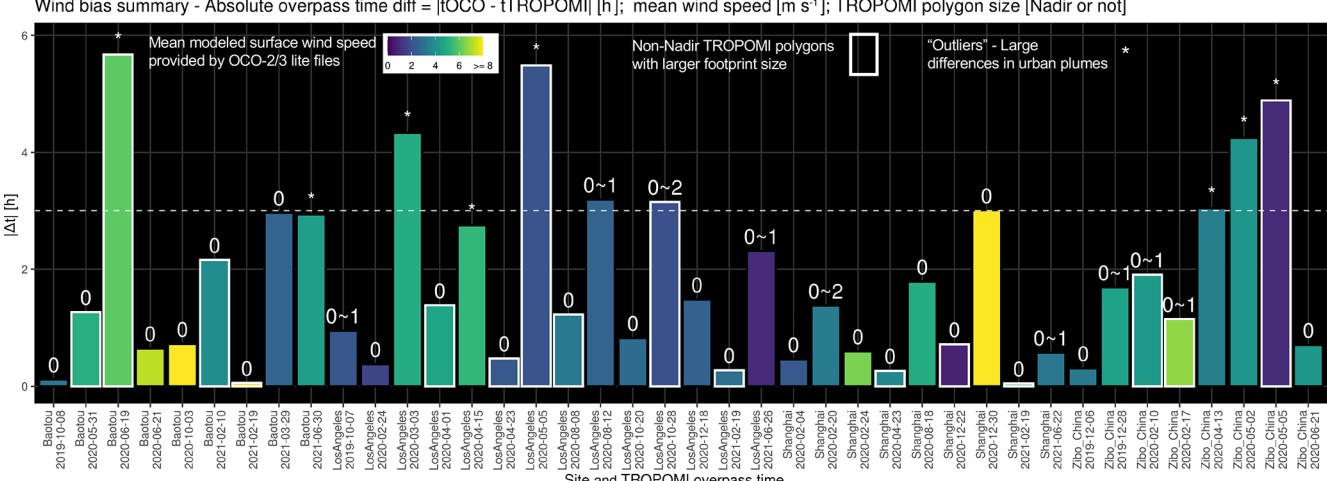

**Figure 8.** A summary of the wind directional shift between OCO-2/3 and TROPOMI overpass times. The $y$ axis denotes the absolute time difference ($|\Delta t|$ in hours) per overpass. The color of the bar represents the instantaneous modeled surface wind speed [m s$^{-1}$] from OCO-2/3 Lite files. Bars labeled with an asterisk indicate that urban plumes between two overpass times differ so much that they cannot be brought into agreement with a simple plume rotation. The number on each bar (e.g., 0–1) denotes the number of TROPOMI polygons that had to be shifted to align the urban plumes at the two times. For example, 0–1 means that TROPOMI polygons over certain locations are shifted by one grid. Bars labeled with a zero on top indicate that a manual plume shift was not required. Bars with white outlines indicate that the sampled TROPOMI soundings on that date were non-nadir with larger pixel sizes.

very few OCO soundings may be located right on the boundaries of TROPOMI polygons. Nevertheless, we find no significant bias associated with the number of OCO soundings per TROPOMI polygon in the heavy industry analysis.

Another factor that we did not explicitly account for is the secondary CO production from both anthropogenic and biogenic VOCs (AVOCs, BVOCs). Under a cascade of reactions in favorable conditions, VOCs emitted from the upwind source location are oxidized to CO at various rates, which may result in higher CO at the downwind sounding location and divergence between enhancement ratios and emission ratios. As BVOCs are usually associated with shorter lifetimes compared to many AVOCs (e.g., Surl et al., 2018), we discuss BVOCs and AVOCs separately. BVOCs can contribute significantly to the total CO source at the regional scale, especially during growing seasons (e.g., Miller et al., 2008; Hudman et al., 2008; Gonzalez et al., 2021). However, since BVOCs, like biogenic $CO_2$, come mainly from rural areas outside the city, subtracting the localized CO background using CO observations outside the urban plume minimizes the impact of BVOCs on the derivation of CO enhancements. The lifetimes of AVOCs are long enough, except for a few species, including alkenes (Surl et al., 2018). Without a good observational constraint on the VOC composition and group-specific emissions for different cities around the globe over the years, it would be challenging to accurately quantify the impact on atmospheric XCO and ER$_{CO}$ of AVOCs emitted from urban areas or specifically from industrial areas. More future efforts regarding urban VOCs may include (1) identifying good proxies that can be measured from space and can

capture the bulk AVOC characteristics (e.g., formaldehyde, Zhu et al., 2014) and (2) interpreting such observations, e.g., by utilizing chemical transport models for source attribution (Gonzalez et al., 2021). Note that the noise/uncertainty in current daily TROPOMI formaldehyde observations may be too large for daily resolved analyses.

## 4.4 Implications for inventory evaluation

This work provides insights into estimating emission ratios from future satellite sensors, as ERs help pinpoint hotspots with poor combustion efficiency, which inform sub-city emission/pollution control efforts.

Satellite-based ER estimates help in the evaluation of sector-specific emission factors and source locations adopted in bottom-up emission inventories (e.g., Silva and Arellano, 2017). Substantial contrast in both the magnitudes and spatial distribution of enhancement ratios can be found between the observations and forward simulations (using X-STILT column footprints and sectoral emissions from EDGARv5). Taking Shanghai as an example, simulated enhancement ratios using prior emissions appear to be much higher ($>$ 50 ppb ppm$^{-1}$, Fig. S13b) than observed ratios (mostly $<$ 30 ppb ppm$^{-1}$, Fig. S13a). Regarding the spatial distribution, simulated enhancement ratios using total FF emissions mimic simulated enhancement ratios using only industry-related emissions (Fig. S13b versus c), and both simulated ratios differ substantially from observed enhancement ratios. Such model–data mismatches may result from inaccurate activity data and emission factors of EDGAR as well as atmo-

spheric transport uncertainties. This preliminary analysis illustrates that satellite observations of trace gases could be used to evaluate emission factors adopted in bottom-up emission inventories. More sophisticated approaches, such as flux inversions (Hedelius et al., 2018; Brioude et al., 2011, 2012; Palmer et al., 2006), may better constrain sector-specific CO and $CO_2$ emissions from inventories.

Spatial proxies, including nightlight data from the Black Marble (https://blackmarble.gsfc.nasa.gov/, last access: TS12) and sophisticated urban land cover datasets, can support not only the development of emission inventories but also sector-orientated evaluations with atmospheric observations of $CO_2$ and co-emitted pollutants. This work demonstrates the benefit of using high-resolution urban land cover classifications to provide independent information about locations of various anthropogenic activities, building structures, and vegetation coverage.

## 5 Conclusions

We investigated fossil fuel combustion efficiency by quantifying the emission ratios of CO and $CO_2$ across Los Angeles, Shanghai, Baotou, and Zibo (Zouping) using nearly coincident observations of TROPOMI XCO and OCO-2/3 $XCO_2$. Multiple swaths of observations collected by OCO-3 SAMs cover a much broader area relative to the OCO-2 swaths, facilitating the determination of background values and the separation of emission signals from different parts of a city. We incorporated spatial gradients in background values by calculating the background per swath and correcting for the urban–background gradient due to non-anthropogenic sources and sinks. Sensor-specific averaging kernel profiles and meteorological conditions were accounted for using an atmospheric transport model (X-STILT). The ratio between XCO and $XCO_2$ enhancements without considering such sensor-specific factors is normally lower than the emission ratio. Cases with severe asynchronicity, specifically those with overpass time differences of over 3 h, correspond to significant changes in urban plumes. Properly accounting for the overpass-specific meteorological conditions or source–receptor relationship and identifying only the soundings influenced by urban emissions is critical when estimating ERs for cities, and is realized using an atmospheric transport model. Our model approach is then used to identify soundings strongly affected by heavy industry. As a result, the overall city-level $ER_{CO}$ for Shanghai ($10.2 \pm 0.4\,\mathrm{mmol\,mol^{-1}}$) is slightly larger than that for Los Angeles ($9.6 \pm 0.5\,\mathrm{mmol\,mol^{-1}}$). Industry-related $ER_{CO}$ values for Shanghai are much larger than its city-level average, whereas industry-related $ER_{CO}$ values for LA are slightly lower than its city-level average. ERs tied to heavy-industry regions in Shanghai ($18.8\,\mathrm{mmol\,mol^{-1}}$) are approximately equal to the city-level $ER_{CO}$ for the industry-orientated city of Baotou ($17.3 \pm 0.5\,\mathrm{mmol\,mol^{-1}}$). High ERs highlight the poor combustion efficiency tied to certain industrial activities, e.g., metal production (Table 1).

Future satellites (e.g., GeoCarb, TEMPO, CO2M) will provide better spatial and temporal coverage of $XCO_2$ and relevant co-located tracer observations, making it possible to monitor and verify temporal trends and variations in the combustion efficiency over hotspots within an urban area, which will provide significant guidance for urban planning and emission control.

## Appendix A: List of prior studies collected in Fig. 1

**Table A1.** A summary table of sector-specific and city-specific emission ratios of CO to $CO_2$ reported in other sources CE3 (including measurement years, locations, paper references, and additional notes). NA = not available.

| Sector in Fig. 1a | Location | Years | Reference | Additional notes |
|---|---|---|---|---|
| Traffic | Denver, US | 1997 | Bradley et al. (2000) | |
| Traffic | Switzerland | 2004 | Vollmer et al. (2007), Table 2 | Gubrist Tunnel |
| Traffic | CONUS | 2005–2007 | Bishop and Stedman (2008) | Chicago, Denver, Los Angeles, Phoenix |
| Traffic | Paris | 2012 | Ammoura et al. (2014), Table 2 | Tunnel (congestion vs. moving) |
| Traffic | Switzerland | 2011 | Popa et al. (2014), Table 1 | Islisberg Tunnel (moving) |
| Traffic | Beijing, China | 2007 | Westerdahl et al. (2009) | Diesel heavy-duty vs. gasoline light-duty |
| Shipping | China | 2011 | Zhang et al. (2016), Table 3 | Diesel engines; estimated from EFs |
| Shipping | Western Europe | 2007 | Moldanová et al. (2009), Table 5 | Diesel engine; estimated from EFs |
| Shipping | Texas | 2006 | Williams et al. (2009), Fig. 2 | Diesel engines; estimated from EFs |
| Biomass burning | Global | NA | Akagi et al. (2011), Tables 1–2 | Estimated from EFs |

| Urban areas in Fig. 1b | Observation years | Reference | | |
|---|---|---|---|---|
| Los Angeles (LA) | 2002 and 2010 | Brioude et al. (2013) | | |
| LA | 2007–2008 | Djuricin et al. (2010) | | |
| LA | 2008 and 2010 | Wennberg et al. (2012) | | |
| LA | 2010 | Silva et al. (2013); Silva and Arellano (2017) | | |
| LA | 2013–2016 | Hedelius et al. (2016) | | |
| LA | 2019–2021 | This study, Fig. 7 | | |
| Pasadena | 2007 | Wennberg et al. (2012), Table 2 | | |
| Sacramento | 2009 | Turnbull et al. (2011a), Sect. 3.2 | | |
| Indianapolis (Indy) | 2012–2014 | Turnbull et al. (2015) | | |
| Salt Lake City (SLC) | 2015–2016 | Bares et al. (2018), Table 2 | | |
| Edinburgh | 2005 | Famulari et al. (2010), Table 1 | | |
| Paris | 2010 | Lopez et al. (2013) | | |
| Paris | 2010–2014 | Ammoura et al. (2016), Table 1 | | |
| London | 2006 | Harrison et al. (2012), Fig. 27 | | |
| London | 2012 | O'Shea et al. (2014), Table 3 | | |
| London | 2016 | Pitt et al. (2019), Table 2 | | |
| Rotterdam | 2011 | Super et al. (2017) | | |
| German Alps | 2012–2013 | Ghasemifard et al. (2019) | | |
| Hungary | 2017 | Haszpra et al. (2019), Table 1 | | |
| St. Petersburg | 2019 | Makarova et al. (2021) | | |
| Miyun | 2004–2008 | Wang et al. (2010), Table 2 | | |
| Beijing | 2006 | Han et al. (2009), Fig. 11 | | |
| Shangdianzi | 2009–2010 | Turnbull et al. (2011b) | | |
| Nanjing | 2011 | Huang et al. (2015), Sect. 3.4.2 | | |
| Seoul | 2016 | Tang et al. (2018), Table 3 | | |
| Seoul | 2019 | Sim et al. (2020), Table 2 | | |
| Jingdezhen | 2017–2018 | Xia et al. (2020), Table 3 | | |
| Beijing | 2019 | Che et al. (2022), Table 2 | | |
| Zibo, Baotou, Shanghai | 2019–2021 | This study, Figs. 6–7 | | |

**Code and data availability.** OCO-3 L2 B10r XCO$_2$ data and TROPOMI XCO data were accessed at https://doi.org/10.22002/D1.2046 (Eldering, 2021) and https://doi.org/10.5270/S5P-1hkp7rp (ESA, 2018), respectively.
X-STILT code has been modified to work with TROPOMI data archived on GitHub branch at https://github.com/uataq/X-STILT (last access: TS13) (DOI: https://doi.org/10.5281/zenodo.2556989, Wu et al., 2019 TS14). We kindly ask users to follow the code policy for utilizing and acknowledging the X-STILT code for
interpreting TROPOMI column data. Hourly NEE fluxes from SMUrF are archived in the Oak Ridge National Lab DAAC at https://doi.org/10.3334/ORNLDAAC/1899 (Wu and Lin, 2021). The urban land cover classification from WUDAPT can be downloaded from https://www.wudapt.org/the-wudapt-portal/ (DOI:
https://doi.org/10.5281/zenodo.6364594, Stewart and Oke, 2012; Demuzere et al., 2022b) TS15.

**Supplement.** The supplement related to this article is available online at: https://doi.org/10.5194/acp-22-1-2022-supplement.

**Author contributions.** DW designed and carried out this analy-
20 sis. JL, POW, and PIP supervised this study. RRN, MK, and AE provided the bias-corrected B10 data for the OCO-3 SAMs used in this work. All authors participated in the interpretation of the results and in the writing and editing of the paper.

**Competing interests.** The contact author has declared that none
of the authors has any competing interests.

**Disclaimer.** Publisher's note: Copernicus Publications remains neutral with regard to jurisdictional claims in published maps and institutional affiliations.

**Acknowledgements.** The computations presented here were
30 conducted in the Resnick High-Performance Computing Center, a facility supported by the Resnick Sustainability Institute at the California Institute of Technology. The first author appreciates the discussion with Joshua Laughner, Eric Kort, Tomohiro Oda, and John Lin. We thank Julia Marshall and the second anonymous ref-
35 eree for their careful reading of our submitted manuscript and for their constructive suggestions that have helped improve our study.

**Financial support.** The production of the OCO-3 science data products used in this paper was carried out at the Jet Propulsion Laboratory, California Institute of Technology, under a con-
40 tract with the National Aeronautics and Space Administration (prime contract number 80NM0018D0004). The research effort was funded by the Jet Propulsion Laboratory Research and Technology Development project R.21.023.106. The analysis was supported by the W. M. Keck Institute for Space Studies and by
45 the National Aeronautics and Space Administration (grant no. 80NSSC21k1064).

**Review statement.** This paper was edited by Jason West and reviewed by Stijn Dellaert and Julia Marshall.

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

## Remarks from the language copy-editor

CE1    "A" inserted. Or do you mean "Modern power generation plants that use... war often.. emmiters"?
CE2    Here as well, "a" inserted.
CE3    Please confirm change from "past literatures" to "other sources".

## Remarks from the typesetter

TS1    Please give an explanation of why this needs to be changed. We have to ask the handling editor for approval. Thanks.
TS2    Please give an explanation of why this needs to be changed. We have to ask the handling editor for approval. Thanks.
TS3    Please confirm citation.
TS4    Please confirm added citation. Concerning the "Global Coal Mine Tracker"-reference: is there more information?
TS5    Please confirm.
TS6    Please give an explanation of why this needs to be changed. We have to ask the handling editor for approval. Thanks.
TS7    Please give an explanation of why this needs to be changed. We have to ask the handling editor for approval. Thanks.
TS8    Please confirm.
TS9    Please confirm.
TS10    Please confirm.
TS11    Please give an explanation of why this needs to be changed. We have to ask the handling editor for approval. Thanks.
TS12    Please note that all URLs need to include a date of last access, because they may expire. Please provide this information.
TS13    Please provide date of last access for the gitHub URL.
TS14    Please confirm citation.
TS15    Please confirm added DOI and citations.
TS16    Please check URL and provide date of last access.
TS17    Please confirm addition of labels.
TS18    Please confirm reference list entry.
TS19    Please confirm article number.
TS20    Please see my note above concerning URLs. Please provide date of last access.
TS21    Please provide DOI.
TS22    Please confirm reference list entry.