# Peer review of "Towards sector-based attribution using intra-city variations in satellite-based emission ratios between CO2 and CO"

_Atmospheric Chemistry and Physics, 2021_

## Referee Comment (RC1)

**Wu et al., 2022: Towards sector-based attribution using intra-city variations in satellitebased emission ratios between CO2 and CO. ACP**

**General comments:**

- Interesting and relevant addition to existing literature.
- Comprehensive description of the methodology, including many relevant figures detailing steps in the process. Clearly a substantial effort was put into the work.
- Discussion of the results (section 3.2) is somewhat limited in depth compared to the rest of the study.
- Good discussion of the limitations. However, one potentially relevant limitation is not covered (see major specific comments).
- Overall, paper is well written. The writing in some sections could still be improved by a thorough read through (e.g. section 4.3).

**Major specific comments:**

The authors combine satellite data on XCO and XCO2 that are collected at different overpass times. The authors have identified 3 factors that may complicate the combination of this data into accurate ER\_CO values, which are discussed at length and properly considered in the data processing and interpretation. However, it appears one potentially relevant factor is missing, namely the timing of the CO and CO2 emissions themselves. While possibly less relevant for industrial sources, other emission sources, such as road transport and residential combustion, may follow a specific temporal pattern where a time difference of 1-3 hours could lead to a mismatch in observed XCO and XCO2 even when wind conditions remain relatively constant. Example: XCO2 is observed during rush hour and XCO two hours later. This could add to the uncertainty in derived ER\_CO values when overpass time differences are larger but remain < 3 hours.</li>

**Minor specific comments:**

**Abstract**

- 15: "After removing those cases..." is the exclusion of cases only on the basis of a t > 3 hours' time difference or also separately on significant wind direction or plume shape changes?
- 18-19: This statement seems incorrect. If I understand correctly, a low ER translates to a higher combustion efficiency (... of heavy industry in LA compared to the city-wide value). In the comparison between cases, it would be preferable to talk either of the combustion efficiency or the ER, and not compare the combustion efficiency with the ER as this may lead to confusion.

**1** Introduction**

37: This statement is a bit tricky. In my opinion, for greenhouse gas emissions, the only key solution would be avoiding combustion of fossil fuels altogether. For some air pollutants such as NOx, higher thermodynamic efficiencies may actually be accompanied with higher emissions. For many air pollutants, abatement technologies such as particulate filters of catalytic converters are more successful in reducing emissions than increasing the combustion efficiency.

**2 Data and methodology**

- Figure 2: "FF XCO2 enhancements...". Have the concentrations in these images indeed been corrected for the background/non-FF sources?
- o 216-217: None of these 3 methods involve prior information on emissions, correct?
- 231-232: How does this method exclude observations elevated by another city? Simply by spatially limiting the plume definition sufficiently to avoid interference from another city?

**3 Results**

- Section 3.1: Here a discussion of a potential interference of changes in the emission of CO2 and CO in the target area between two overpasses that may be up to 3 hours apart would seem appropriate (see major comments).
- 319-320: Not clear why the effect of biogenic and pyrogenic contributions itself is limited by removing overpasses interfered by wind shift.
- 348-349: Urban-background gradients in biogenic anomalies would not change FF (fossil fuel) enhancement by definition.
- 382: While it is clear for the enhancements, why the lockdown should influence ER\_CO in a specific direction is not directly clear. Perhaps comment on a potential mechanism here. Also, for Zibo the enhancements on 2020-02-11 appear larger than on 2019-12-28.
- 392-394: "In contrast to LA, ...". It is suggested here that wind speeds and direction do not change dramatically for LA, while the opposite appears to be suggested in line 386.
- 425: This is an important point for interpreting the results in terms of "combustion efficiency". While in many industrial processes, hydrocarbons are combusted with the intention of achieving as high as possible combustion efficiencies, in iron/steel production, CO actually serves an important function during the process. While an iron/steel plant will aim at burning as much as possible the CO before release to the atmosphere, it still is a tricky comparison with other sectors under the concept of 'combustion efficiency'. Perhaps a short mention of this would be appropriate.

**4 Discussion**

- 459-460: "...constraint can be relaxed". This statement may need to be reconsidered when also taking into account the changes in emissions of (FF) CO2 and CO during the day (see major comments).
- 461-462: This sentence is not clear to me. The text appears to suggest that the plumes during the overpasses on May 31, 2020 are sufficiently similar. Still, in figure 8 it is marked as an outlier (\*). This also makes the relation to the next sentence (461-462) unclear.
- 497: I had to reread the paragraph on VOC's several times and I'm still not really sure what is the conclusion on the relevance for the present work. Please consider guiding the reader a bit more.
- 516: "using the industry emissions from EDGAR". Does this comparison indeed only consider industry emissions or also the other sectors (e.g. those shown in Fig S11d-f)?

**Supplementary information**

• Figure S7: The figure does not show the overpass time, that could for example explain the difference between the lower right figures (Zibo 2020-05-05 and Zibo 2020-06-21). Perhaps the colour can be used to indicate overpass time instead of the month (which is already shown above each graph).

**Technical corrections:**

**Abstract**

• 3: Consider using "combustion efficiency" consistently instead of "burning efficiency".

**1** Introduction**

- o 43-45: "Benefit ... reported". Please check the sentence structure, it is not fully clear.
- o 50: "be difficult." some words are missing. Perhaps "...be difficult to identify"?
- 56: "diagnosed". consider using a different term, e.g. "analysed".
- Figure 1:
  - What sector(s) is "biofuel + ff" referring to? All anthropogenic combustion sectors?
  - In fig.1b the vertical (minor) grid lines fall in between years. The figure may be easier to read if these lines coincide with the years.
- 65-66: Not clear what is meant by "...the generalization and representation of gridded ERs."
  Also, the next few sentences could be written a bit more clearly.

**2 Data and methodology**

- 107: "and surrounded coal-fired" —> "surrounding"? or "presence of multiple coal-fired power plants"
- 109: "that is the spatial extent" -> "that is in the spatial extent"
- 111: "that estimated" —> "that are estimated"
- 111: "FF enhancements" this term is a bit vague.
- 116: "we illustrate how much ERs ... can be extracted." What does this mean?
- 124: "north hemisphere". This term appears to be more commonly spelled as "northern hemisphere".
- 163: "interfere the calculation" —> "interfere with the calculation"
- 218: "swaths of stretching" —> "swaths stretching"
- 220: "have been emphasized" —> "has been emphasized"

**3 Results**

o 375: No need to present two decimals for the crude iron production.

**4** Discussion**

- 451: "considered for when"
- 456: "Interfering the" —> interfering with the".
- 456-457: Please rephrase, as the lack of concurrence is not itself inducing the wind directional shift.
- o 464: "future geostationary satellite" singular or plural?
- 464: What is meant by "spontaneously" here?
- 468: "a urban" —> "an urban"
- 512: "efficiency can" —> "efficiency, which can"
- 513-514, 521: "inventory" -> "inventories"
- Section 4.3. the readability of the text in this section could be improved by a thorough readthrough

**5** Conclusion**

545: "industry-related ER\_CO slightly lower" —> "industry-related ER\_CO are slightly lower"

**Supplementary information**

• Figure S10: "distribution is stewed towards" —> "distribution is skewed towards"

---

## Referee Comment (RC2)

The study presents a technically detailed implementation of a Lagrangian model adapted for use with column data, X-STILT, focusing on emission ratios of CO:CO2 on sub-city scales using XCO measurements from TROPOMI and XCO2 measurements from OCO-2 and OCO-3. The level of technical detail is quite good (with a few exceptions, see comments below), and it is nice to see that the adapted X-STILT code has been made publicly available. The approach is interesting, particularly in its attempt to not make use of emissions inventories as part of the prior information. However some potential sources of error have been overlooked, particularly with respect to the high-frequency variability of the CO:CO2 ratio, which seems to be assumed to be constant during the time between measurements. This is not something that the current study can really correct, but it needs to be included in the discussion as a clear limitation of the current results. Despite these limitations, the study is certainly appropriate for publication in ACP once these concerns are addressed. A good proof-reading of the paper is also needed before resubmission: while it was almost always clear what was meant, and the paper was well-written and easy to follow, there were lots of missing articles etc. that the native speakers among the co-authors could clear up quickly.

**Major concerns:**

The authors have gone to great pains to try to correct for temporal shifts between the measurements, considering the impact of the different meteorological conditions and averaging kernels. What was not taken into account is changes in the CO:CO2 ratio over the course of a day. While this may not be as critical for heavy industry and power generation, other sectors (e.g. traffic) have highly heterogenous emission ratios in time, depending also on traffic patterns. Having the XCO and XCO2 measurements offset by even a couple of hours complicates this approach considerably, and might also cloud the proposed analysis of temporal trends in the emission ratio over the year. That does not mean that nothing can be learnt from this approach, only that this neglected error source needs to be explicitly described. In any case, GeoCarb data will make such analyses considerably easier in the future.

**Minor concerns:**

What the authors mean by "FF" needs to be made clear. The implication is that the emission signature of fossil fuels is being measured directly, which is clearly not the case. Emission from combustion, sure. There is no capacity to separate e.g. biofuel from fossil fuels in this approach, and this needs to be made clear.

L37: While this is true for some air pollutants such as CO, this is not true for many CO2. In fact, increasing efficiency during combustion activities increases the amount of CO2 and NOx emitted (unless the latter is scrubbed) (e.g. Lama et al., 2020). The key to reducing emissions of greenhouse gases is to reduce combustion, period.

L223-224: Why use a rectangle as the source function? Was this to be "independent of emission inventories", while still knowing that emissions are locating within the city?

L231: Regarding the second point about excluding observations elevated by another city: yes, this excludes enhancements outside the model-defined urban plume, but what about enhancements from other cities that might also be contributing to enhancements within the urban plume? This seems to be the case for the XCO values shown in Figure 3c. Is there any way to correct for these values? Especially given that there do not seem to be XCO2 values over the same area?

L235: Is there any way to mark on here which soundings were used to define the background? Some more quantification of the information would also be useful here. Which latitude range was used to define the background for the OCO-2 swath? In Figure 3a I don't see any soundings outside the urban

plume region... For 3b it is somewhat clearer, aided by the coastal cut-off (although point sources near the coast are clearly in the "background", but how the area would be defined for Figure 3c is really vague. With this level of detail, the approach would be hard to replicate.

Section 2.2.4: It's unclear how the observation uncertainty is computed here. Each sounding has a reported uncertainty that comes with the data product – is this the retrieval error? The "measurement noise" seems to be equated with the standard deviation of the retrievals within a TROPOMI footprint. But this isn't really what is usually meant by measurement noise. It is not completely clear how these two types of errors (one per OCO-2 sounding and one per larger TROPOMI sampling) are combined, only that the retrieval errors are aggregated in a "standard-deviation-of-mean" manner, which sounds like it's being divided by the square root of the number of soundings. This assumes that the measurements and their errors are independent of one another, which is not the case. This almost certainly underestimates the measurement uncertainty on the X\_{ffCO2} term. A formula here is certainly needed.

Technical/language comments:

L43-45: I'm confused by this sentence. Maybe should "Benefit" be "Benefiting"? In any case, the sentence should be rewritten to make it clearer, or even split into two sentences, starting with the second half, i.e. something like "The ratio... between tracers is reported. This has the benefit that errors ... cancel out."

L46: would remove "their"

L50: ...difficult to detect?

L53-54: Rewrite this, something like: "Given its much longer lifetime, CO is much easier to interpret..." The "on the other hand" doesn't fit here as written.

Figure 1 caption: "The x-axis indicates..."

L82: such gradient -> either "such a gradient" or "this gradient"

L87-88: in vertical -> in the vertical

L89: by the gaps in prior literature -> by gaps in the existing literature

L92: adopt or adapt?

L97: quantify accurately -> accurately quantify

L101: implication and limitation -> implications and limitations

L107: surrounded -> surrounding

L109: urban plume that is the spatial extent -> urban plume, defined as the area

L111: that estimated -> that are estimated

L113: requires estimate -> requires an estimate

L116: would remove "much"

L155: accounted for AK -> accounted for the AKs?

L156: As result -> As a result

L160: evaluations -> evaluation

L163: interfere -> interfere with; also: more explanations in Sect. -> for more explanation, see Sect.

Figure 2 caption: The citation for the Google Maps data in the last sentence doesn't sound quite right – please check what it's supposed to be (i.e. adopted the Google Maps what?)

L166: and atmospheric transport model -> and an atmospheric transport model

L175: for sounding-specific -> for the sounding-specific

L178-181: I think I understand what is meant here, but it's a bit hard to parse. When I hear "pathways" I'm thinking of chemical reactions, and I'm not sure what is meant by an air parcel being "tied" to somewhere, or correction terms being "attached" (perhaps "applied" would fit better)? In any case this should be rewritten for clarity.

L193: corresponding for -> corresponding to

L196: wind condition -> wind conditions

L200: by sounding -> by the sounding

L201: Because AK -> Because the AK

L204: If -> By

L208-209: I guess the meteorological conditions and the AK profile are specific to the sounding, not just the sensor?

L209: condition -> conditions

L216: combines -> combine

L216-217: This seems a bit backwards – isn't the first method more independent of information about emissions, unlike the two modelling-based approaches?

L218-219: improve it over what?

L222: soundings within -> soundings as within

L258-262: I would split this into two sentences.

L265: lasts for -> ranges from

L267: are -> is

L273: Observed uncertainty of XCO2 are -> Observation uncertainty of XCO2 is

Figure 4: Please add a coastal outline in panels c and f to make it easier to interpret. (Masking water would also be an option.)

Figure 4 caption: lightgray should be two words.

L301: I think a word is missing. Maybe: "Those industry coverage maps are then convolved..."

L311: I would remove "originating" here, it is more confusing than helpful.

L313: When you write "too low valid soundings" do you mean "too few valid soundings" or "too low enhancements"?

L313-314: remove "the few"

L319: interfered by wind shift -> affected by shifting winds

L329-330: Recommended change: "Again, the colored contours and curves in Fig. 5 indicate neither the intensity of concentrations nor flux fields (as no prior emissions are used), but rather the likelihood of urban plumes determined solely by atmospheric dispersion."

L332: but problematic -> but becomes problematic

L335: remove "cases"

Figure 5: in figure label, it should be "caution" rather than "cautions"

L364: I would suggest replacing "atmospheric movement" with "transport". Also, something seems to have gone wrong with many of the subscripts in this paragraph (e.g. ECO instead of E\$\_{CO}\$ in LaTex syntax).

Figure 6 caption: The second sentence and the second-last sentence seem to repeat the same information. Remove one? Also, it is mentioned here that only regression lines with positive slopes were chosen from the Monte Carlo experiment. What proportion of these lines needed to be removed?

Figure 6: the \gamma bio term shown on the plots should be in units of ppm, right? Also, is this somehow different from the \delta X\_{bio} discussed previously? If not, please make this consistent.

L385: remove second comma, also change "them" to "these overpasses".

L392: in time -> in the time

L406: tend -> tends

L408 (and elsewhere): perhaps "industry-dominated" might be more appropriate in some places than "industry-dominant"?

L432: i.e., less -> i.e., those less

L452: the Solar-Induced -> remove "the"

L455: function -> functional

L456: either "interfering" -> "interfering with" or "interfering" -> "affecting"

L456: "wind directional shift induced by" -> "the shift in wind direction due to"

L464: spontaneously -> simultaneously

L464: do you mean "a future geostationary satellite (i.e. GeoCarb)" or "future geostationary satellites"?

Figure 8 caption: Text could use a bit of work. Suggestion: "indicate the urban plumes between two times differ significantly that a simple plume rotation fails to fix" -> "indicate that that urban plumes between the two overpass times differ so much that they cannot be brought into agreement with a simple plume rotation". Also: what is the meaning of  $0^2$  and  $0^2$ ? Are they shifted or not?

L479: begun -> began

L483: take -> takes

L492: to coarser -> to the coarser

L494: may locate -> may be located

L500: against OH -> against the OH

L503: contributed to 21.2% -> contributed 21.2%

L503: but negligible -> but a negligible

L504: season, the -> season, and as such the

L504: remove "likely", "encapsulated" -> "included", "yield" -> "have"

L506-507: "...whether ... remains unclear" -> "it is unclear whether AVOCs... of interest."

L512: "can help" -> "to"

L513-514, and L521: inventory -> inventories

L515: footprint -> footprints

L520:

L527: in informing locations of -> to provide information about

Supplement:

Figure S1 caption: average -> averaged

Figure S3 caption: unique to each satellite sounding given unique -> unique to each satellite sounding, giving a unique

Figure S3 caption: Column footprint -> The column footprint

Figure S4 caption: "these resultant normalized fraction Pind(x, y) informs the influence on the observation at a given sounding (white rectangle) due to heavy industry. Lighter the color, stronger impact from heavy industry in LA." -> "these resultant normalized fractions Pind(x, y) show the influence of heavy industry on a given sounding (white rectangle). The lighter the color, the stronger impact from heavy industry."

Figure S5 caption: "of urban-background" -> "of the urban-background"; "Since biogenic" -> "Since the biogenic"; "Two sets of footprint totals" -> "The two sets of footprints"; "difference in AK" -> "difference in the AKs"; "Higher the footprint ratio, larger the discrepancies" -> "The higher the footprint ratio, the larger the discrepancy"

Figure S6 caption: "during two" -> "during the two"

Figure S7 (and others): Is the date format in the plots (YYYYMMDD) consistent with the Copernicus guidelines?

Figure S7 caption: "close to the noon, daytime carbon sink dominant leading" -> "close to noon, the daytime carbon sink dominates, leading"; "nighttime respiration dominant," -> "respiration dominates the biogenic fluxes," (I would remove "nighttime" because winter cases are also mentioned.)

Figure S9 caption: "with or without the account of the urban-rural biogenic gradient over Zibo on June 21, 2020" -> "over Zibo on June 21, 2020, with and without taking the urban-rural biogenic

gradient into account"; "light grey shading denote" -> "light grey shading denotes"; "Such positive gradient" -> "Such positive gradients"

Figure S10 title: A bit awkward, would suggest "Log-normal distributions of ERCO [ppb-CO / ppm/CO2]

Figure S10 caption: "on log-normal" -> "on a log-normal"; "stewed" -> "skewed"; Also, aren't the values skewed towards the higher end (i.e. positive skew or right-skewed) if the mean is higher than the median?

Figure S11: Should the title for panels c) through f) be Sectoral CR\_CO or ER\_CO? Also, I guess panel f) should be marked "residential" rather than "resident"? (Perhaps "road traffic" might also be better for d)...)

Reference in the review:

Lama, S., Houweling, S., Boersma, K. F., Eskes, H., Aben, I., Denier van der Gon, H. A. C., Krol, M. C., Dolman, H., Borsdorff, T., and Lorente, A.: Quantifying burning efficiency in megacities using the NO2/CO ratio from the Tropospheric Monitoring Instrument (TROPOMI), Atmos. Chem. Phys., 20, 10295–10310, https://doi.org/10.5194/acp-20-10295-2020, 2020.

---

## Author Comment (AC1)

**Point-to-point responses**

We appreciate the careful read-through and constructive comments from both reviewers. The reviewers' comments are copied in **black** with our point-to-point responses in **blue** and revised text in **red** or pasted.

**Response to reviewer #1:**

**General comments:**

- Interesting and relevant addition to existing literature.
- Comprehensive description of the methodology, including many relevant figures detailing steps in the process. Clearly a substantial effort was put into the work.
- Discussion of the results (section 3.2) is somewhat limited in depth compared to the rest of the study.
- Good discussion of the limitations. However, one potentially relevant limitation is not covered (see major specific comments).
- Overall, paper is well written. The writing in some sections could still be improved by a thorough read through (e.g. section 4.3).

We truly thank reviewer #1 for the recognition and constructive comments and attempted to revise the manuscript by
1) adding sensitivity analyses to examine how changes in hourly emissions between OCO-3 and TROPOMI overpass times may affect the enhancement ratio. We now analyze the column data from TCCON at Caltech
2) clarifying the relevant text to avoid confusion, especially the important point on the interpretation of combustion efficiency that varies significantly with regions, technologies, and the type of activities.

**Major specific comments:**

The authors combine satellite data on XCO and XCO2 that are collected at different overpass times. The authors have identified 3 factors that may complicate the combination of this data into accurate $ER_{CO}$ values, which are discussed at length and properly considered in the data processing and interpretation. *However, it appears one potentially relevant factor is missing, namely the timing of the CO and CO2 emissions themselves. While possibly less relevant for industrial sources, other emission sources, such as road transport and residential combustion, may follow a specific temporal pattern where a time difference of 1- 3 hours could lead to a mismatch in observed XCO and XCO2 even when wind conditions remain relatively constant.* Example: XCO2 is observed during rush hour and XCO two hours later. This could add to the uncertainty in derived $ER_{CO}$ values when overpass time differences are larger but remain < 3 hours.

We agree with reviewer #1 that the mismatch in the timing of CO or $CO_2$ emissions themselves may influence the $ER_{CO}$, although the overpass time difference has been limited to 1-3 hours. One can likely rely on bottom-up emission inventories or prior assumptions to infer hourly emission patterns. However, because emissions or combustion characteristics are what we are solving for, it is quite challenging to properly account for such mismatch WITHOUT involving prior assumptions towards emissions themselves. Such hourly emission variations depend on the relative contributions from individual emission sectors for a given city.

We have now carried out a sensitivity analysis to investigate how such a mismatch in emission timing may affect the $ER_{CO}$, or more generally the high-frequency variability of ERCO (as reviewer #2 also mentioned). Thanks to the TCCON network that provides high-frequency XCO and $XCO_2$ measurements (TCCON 2022). We utilized the latest GGG2020 version with several upgrades including a much-improved prior profile. Because XCO and $XCO_2$ are simultaneously retrieved, we can assume that atmospheric transport associated with two species are the same and their enhancement ratio fully reflected the emission characteristics once species-specific averaging kernel profiles are corrected for following Appendix A2 in Hedelius et al. (2018).

**Here we reported the estimated enhancement ratio at Caltech with background observations from another TCCON site outside the LA basin (see figure and figure caption shown below). Because the temporal frequency of the Caltech TCCON site may not perfectly match that of the background TCCON site, we first averaged observations from both sites to each 20-min interval and calculated the enhancements and enhancement ratio. Indeed - the variation in $ER_{CO}$ can be large throughout the day. Note that such variations may reflect not only the change in FF emissions (e.g., due to traffic) but also measurement noise (given these high-frequency data) as well as changes in the meteorology throughout the day (ocean versus mountain flow for LA). Nevertheless, changes in $ER_{CO}$ between OCO-3 and TROPOMI times inferred from concurrent TCCON measurements appear to be small.**

[Figure]

**Fig. S8** - Time series of observed ERCO at the California Institute of Technology (Caltech) TCCON site (Wennberg et al. 2017) on the OCO-3 overpass dates till June 2021. The background is defined using the NASA Armstrong Flight Research Center (AFRC) site near Lancaster, California (Iraci et al., 2022). Column enhancements with corrections of TCCON averaging kernel are calculated following Hedelius et al. (2018). The overpasses that went into the final result are shown in solid black dots, while the remaining overpasses with significant plume shift between OCO-3 and TROPOMI overpass times are shown in black crosses. The vertical lines indicate the OCO-3 (green) or TROPOMI (red) overpass times in UTC times. The day of the week for each date is shown as follows: Feb 24, 2020 (Mon), Mar 3 (Tues), Apr 15 (Wed), Apr 23 (Thurs), May

5 (Tues), Aug 8 (Sat), Aug 12 (Wed), Oct 20 (Tues), Oct 28 (Wed), Dec 18 (Fri), Feb 19, 2021 (Fri), and June 26, 2021 (Sat). Note that no qualified data exists during the overpass time of OCO-3 or TROPOMI on April 1, 2020. The TCCON data were obtained from the TCCON Data Archive hosted by CaltechDATA at https://tccondata.org. We thank Laura T. Iraci and Coleen M. Roehl for preparing the TCCON data for these two sites.

Even though many more analyses related to the hourly or seasonal $ER_{CO}$ can be drawn from these TCCON observations, they are out of the main goal of the current manuscript in using space-based observations. Since these TCCON and spatially resolved satellite observations (OCO-3-TROPOMI) may have a different emphasis on the emission signals within the LA basin, we tried to avoid unnecessary direct comparisons of their derived $ER_{CO}$ and decided to leave the analysis in the supplementary material but added some discussions in

**Sect. 2.1.3:**

3. *Overpass times, meteorological conditions, and emission variations:* As a result of the overpass time difference between
160    sensors, variations in meteorological conditions (e.g., wind direction and speed) can lead to changes in urban plume shapes detected by the two sensors as they pass by. We dealt with changes in wind speed and wind direction separately. The former is resolved by the "scaling factor" inferred from an atmospheric transport model and the latter undergoes manual evaluations (**Sect. 3.1**). Besides, CO and $CO_2$ emissions themselves can vary over the course of a day, e.g., driven by road transportation and residential sectors. Given the overpass time difference between sensors, it is likely that such
165    mismatch in the timing of CO versus $CO_2$ emissions may affect the observed $ER_{CO}$.

**Sect. 3.1:**

Besides changes in wind directions, CO and $CO_2$ emissions themselves can vary across daytime hours, likely driven by the road transportation and residential sectors. As a result, variations in the derived $ER_{CO}$ across multiple overpasses may reflect not only the variation in combustion efficiencies but also the mismatch in the emission timing. LA may be one of the cities with more distinct daytime changes in emissions compared to industry-centered cities. Fortunately, based on a supple-
355    mentary sensitivity analysis using measurements from the Total Carbon Column Observing Network in Pasadena (TCCON, Wennberg et al., 2017), by limiting satellite overpasses to those with a smaller time difference, $ER_{CO}$ appear to be less variable (**Supplementary Fig. S8**). Future geostationary satellite monitoring $NO_x$ (e.g., TEMPO, Chance et al., 2022) may provide better guidance towards the hourly pattern in urban emissions, especially from the traffic sector with more daytime fluctuations, which have been discovered using surface monitoring networks (e.g., over Chicago; de Foy, 2018).

**And Sect 4.1:**

"If TROPOMI pixel sizes are relatively large (i.e., non-nadir observations) or the wind is steadier, this dt constraint may be relaxed, as long as emissions for a specific city is less driven by sectors with noticeable diurnal cycle (e.g., road transportation)."

**Reference:**
Hedelius, J. K., Liu, J., Oda, T., Maksyutov, S., Roehl, C. M., Iraci, L. T., Podolske, J. R., Hillyard, P. W., Liang, J., Gurney, K. R., Wunch, D., and Wennberg, P. O.: Southern California megacity CO2, CH4, and CO flux estimates using ground- and space-based remote sensing and a Lagrangian model, Atmos. Chem. Phys., 18, 16271–16291, https://doi.org/10.5194/acp-18-16271-2018, 2018.

Wennberg, P. O., D. Wunch, C. Roehl , J.-F. Blavier, G. C. Toon, N. Allen. 2017. TCCON data from California Institute of Technology, Pasadena, California, USA, Release GGG2020R0. TCCON data

archive, hosted by CaltechDATA, California Institute of Technology, Pasadena, CA, U.S.A. https://doi.org/10.14291/tccon.ggg2020.pasadena01.R0

Iraci, L., J. Podolske, C. Roehl, P. O. Wennberg, J.-F. Blavier, N. Allen, D. Wunch, G. Osterman. 2022. TCCON data from Armstrong Flight Research Center, Edwards, CA, USA, Release GGG2020R0. TCCON data archive, hosted by CaltechDATA, California Institute of Technology, Pasadena, CA, U.S.A. https://doi.org/10.14291/tccon.ggg2020.edwards01.R0

**Minor specific comments:**

**Abstract**

15: "After removing those cases..." is the exclusion of cases only on the basis of a t > 3 hours' time difference or also separately on significant wind direction or plume shape changes?

We examined the plume shape for every overpass (e.g., Figure 5) and remove those whose plume shapes differ significantly (i.e., those overpasses with an asterisk in Figure 8). Thus, it is possible that cases with $dt$ > 3 hours are included in the final results because the plume shift is negligible compared to the TROPOMI footprint size. A time difference of 3 hours is summarized here to provide a very conservative threshold if one would like to examine OCO-3 and TROPOMI overpasses for other places.

18-19: This statement seems incorrect. If I understand correctly, a low ER translates to a higher combustion efficiency (… of heavy industry in LA compared to the city-wide value). In the comparison between cases, it would be preferable to talk either of the combustion efficiency or the ER, and not compare the combustion efficiency with the ER as this may lead to confusion.

We apologize for the careless mistake and now just use ERCO when comparing ERs from heavy industry versus the whole city. Right - lower ERCO corresponds to higher combustion efficiency. We attached the relevant text here:
"Results suggest that $ER_{CO}$ impacted by the heavy industry in Los Angeles is slightly lower than the overall city-wide value (< 10 ppb-CO / ppm-$CO_2$)."

**1 Introduction**

37: This statement is a bit tricky. In my opinion, for greenhouse gas emissions, the only key solution would be avoiding combustion of fossil fuels altogether. For some air pollutants such as NOx, higher thermodynamic efficiencies may actually be accompanied with higher emissions. For many air pollutants, abatement technologies such as particulate filters of catalytic converters are more successful in reducing emissions than increasing the combustion efficiency.

We agree with the reviewer and have now revised the relevant text as follows:
"Given the co-benefits of GHG reduction and improved air quality at various scales (Zhang et al., 2017), controlling the consumption of fossil fuels altogether is the key."

**2 Data and methodology**

Figure 2: "FF $XCO_2$ enhancements...". Have the concentrations in these images indeed been corrected for the background/non-FF sources?

The constant background values like Xbg$_{co2}$ and Xbg$_{co}$ in Equations 1 & 2 were subtracted from the total columns, but not the gradients from non-FF sources (i.e., delta X-terms). XCO$_{2.ff}$ enhancements with corrections for non-FF sources were displayed in **Fig. S5a**. To improve the consistency, we have now changed **Fig. 2** to further exclude the non-FF gradient following Equations 1 & 2 and added the XCO$_{2.ff}$ with or without non-FF sources in **Fig. S5** to show the differences. Here are the updated figures for LA on Feb 24, 2020.

**Fig. 2**

[Figure]

a) XCO$_{2.ff}$ [ppm] at OCO-3 scale on Feb 24, 2020, at 19:59 UTC

b) XCO$_{2.ff}$ [ppm] with non-FF source correction at TROPOMI scale on Feb 24, 2020, at 19:59 UTC

c) XCO$_{ff}$ [ppb] from TROPOMI on Feb 24, 2020, at 20:23 UTC

**Fig. S5** In general, XCO$_{2.ff}$ within the urban plume with biospheric corrections appears to be slightly smaller than those without the biospheric corrections, as explained in the main text.

[Figure]

a) XCO$_{2.ff}$ [ppm] WITHOUT non-FF source correction at TROPOMI scale on Feb 24, 2020, at 19:59 UTC

b) XCO$_{2.ff}$ [ppm] WITH non-FF source correction at TROPOMI scale on Feb 24, 2020, at 19:59 UTC

c) Footprint ratio ( = Foot$_{co2}$ / Foot$_{co}$)

216-217: None of these 3 methods involve prior information on emissions, correct?

Correct - this statement has now been removed. For clarification,
- The first statistical method utilized standard deviation or percentiles to derive the background purely from observed XCO$_2$. So, no use of prior emissions.

- The second method uses modeled initial conditions from atmospheric transport models and global $CO_2$ fields. Global $CO_2$ fields may involve prior assumptions on emissions.
- The third method first considered atmospheric transport and identified the background region (but not too far from the target city) and then calculated the median of the observed $XCO_2$ in the background region as the background. An assumption on the rough spatial extent of the city emission (e.g., a rectangle).

231-232: How does this method exclude observations elevated by another city? Simply by spatially limiting the plume definition sufficiently to avoid interference from another city?

This is a good point. Yes - the spatial limiting of the plume domain is realized by the particle distribution and the normalized kernel density function is sufficient to avoid interference from another city in most cases. Also, we only selected the soundings over regions from which the wind is blowing (e.g., to the south or east). With the median statistics for the background value, high XCO or $XCO_2$ values affected by a potential nearby city would play a minor impact on the background.

It is possible to conduct manual checks by releasing particles from a nearby city (e.g., the city of Nantong to the north of Shanghai) and outlining a similar urban plume for a nearby city. Alternatively, if several cities are so lumped together (e.g., three major cities in the Pearl River Delta in China, Ye et al., 2020), the entire metropolitan area can be treated as a whole entity for enhancement or emission estimates. Nevertheless, we may argue that the current approach is sufficient to minimize the influence of background from another city.

[Figure]

**Ref:** Ye, X., Lauvaux, T., Kort, E. A., Oda, T., Feng, S., Lin, J. C., Yang, E. G., and Wu, D.: Constraining Fossil Fuel CO2 Emissions From Urban Area Using OCO-2 Observations of Total Column CO2, Journal of Geophysical Research: Atmospheres, 125, e2019JD030 528, 2020.

**3 Results**

Section 3.1: Here a discussion of a potential interference of changes in the emission of CO2 and CO in the target area between two overpasses that may be up to 3 hours apart would seem appropriate (see major comments).

As explained above, we have added an additional analysis and discussion in Sect. 3.1 (also pasted below) to investigate the impact of changes in hourly emission of $CO_2$ and CO onto observed $ER_{CO}$

> Besides changes in wind directions, CO and $CO_2$ emissions themselves can vary across daytime hours, likely driven by the road transportation and residential sectors. As a result, variations in the derived $ER_{CO}$ across multiple overpasses may reflect not only the variation in combustion efficiencies but also the mismatch in the emission timing. LA may be one of the cities with more distinct daytime changes in emissions compared to industry-centered cities. Fortunately, based on a supple-
> 355  mentary sensitivity analysis using measurements from the Total Carbon Column Observing Network in Pasadena (TCCON, Wennberg et al., 2017), by limiting satellite overpasses to those with a smaller time difference, $ER_{CO}$ appear to be less variable (**Supplementary Fig. S8**). Future geostationary satellite monitoring $NO_x$ (e.g., TEMPO, Chance et al., 2022) may provide bet- ter guidance towards the hourly pattern in urban emissions, especially from the traffic sector with more daytime fluctuations, which have been discovered using surface monitoring networks (e.g., over Chicago; de Foy, 2018).

319-320: Not clear why the effect of biogenic and pyrogenic contributions itself is limited by removing overpasses interfered by wind shift.

We apologize for the confusing text that sounds like we suggested certain causality between the two. The effect of non-FF contributions will not be minimized by removing overpasses with significant plume shifts. The non-FF contributions were only estimated for tracks with insignificant plume shits, which turns out to be minimal (but may not be small for all overpasses and all cities). To clarify, we have modified the text to be:
> "For the final 24 overpasses we selected, temporal variations in the emission pattern and urban-background gradients in biogenic/pyrogenic contributions play minor roles in overpass- or city-level ERs."

348-349: Urban-background gradients in biogenic anomalies would not change FF (fossil fuel) enhancement by definition.

Right - the non-FF sources (namely their urban-background gradients) will not alter FF enhancements since FF enhancements have already been accounted for (Eqs. 1&2). We just would like to clarify that our definition for the background is quite unique, that is a constant value for a given satellite swath. Such *constant background* is derived based on observations of soundings often over the rural areas. While using such constant background to derive the enhancements may fail to account for the urban-rural or urban-background gradients in non-FF sources. We have now revised the text as:
> "As explained in **Sect. 2.2.2**, urban-background gradients in these biogenic anomalies (i.e., $\delta X_{bio}$) were used to correct the constant background $X_{bg}$ (Eq. 1)."

$$\langle E_{CO_2,s} \rangle = \frac{X_{ffCO_2,s}}{\langle XF_{CO_2,s} \rangle} = \frac{X_{obsCO_2,s} - X_{bgCO_2} - \delta X_{bioCO_2,s} - \delta X_{bbCO_2,s}}{\iint XF_{CO_2,s}(x,y)\,dx\,dy}. \tag{1}$$

$$\langle E_{CO,s} \rangle = \frac{X_{ffCO,s}}{\langle XF_{CO,s} \rangle} = \frac{X_{obsCO,s} - X_{bgCO} - \delta X_{bbCO,s}}{\iint XF_{CO,s}(x,y)\,dx\,dy}. \tag{2}$$

382: While it is clear for the enhancements, why the lockdown should influence $ER_{CO}$ in a specific direction is not directly clear. Perhaps comment on a potential mechanism here. Also, for Zibo the enhancements on 2020-02-11 appear larger than on 2019-12-28.

These are good questions. According to the new figure shown below, it seems that XCO2 enhancements for 2020-02-10 are larger than those for 2019-12-28, while XCO enhancements for 2020-02-10 are generally smaller than those for 2019-12-28. The wind directions between two cases are very similar (implied by the urban plumes in solid black curves). The much larger XCO2 enhancements for 2020-02-10 may be due to differences in sampling locations (e.g., 2019-12-28 case is farther away from the city source). Additionally, because 2019-12-28 is not a SAM and there are some missing TROPOMI polygons, only a few TROPOMI polygons (that overlap with the narrow swath of OCO) are used for calculating $ER_{CO}$. Thus, $ECO_2$ in **Fig. 6** for 2020-02-10 appears to be larger than that for 2019-12-28.

[Figure]

Considering differences in the satellite sampling (i.e., locations and available numbers of soundings), we may argue that evaluating $ER_{CO}$ between overpasses would be more robust in evaluating FF enhancements between overpasses. This is also true for our meteorology-normalized emissions (E = XFF / XFootprint) since the model-based meteorology (X-STILT footprint) may be associated with transport uncertainties.

We suspect the drop and then rise in $ER_{CO}$ is tied to the power generation and metal industry around Zibo. Zibo with its neighborhood city Zouping is responsible for ⅛ of the total coal consumption of its entire Shandong province in 2017. According to the global power plant dataset (WRI, 2018, new **Table 1** showen below), the total power generation capacity is around 9720 mW (100% fueled by coal), which is more likely to support the local metal industry (e.g., electrolytic aluminum and iron and steel), as the city itself is small and not heavily populated. Weiqiao Pioneering Group locating in the area has become the world's largest aluminum producer but "used relatively inefficient subcritical steam generators" (https://chinadialogue.net/en/energy/10040-coal-power-and-privilege-china-s-problem-with-industry-owned-generators/). Combining all these info, we suspect that those industries and the nearby power plants may be partially shut down during the COVID lockdown in 2020, which correspond to lower $ER_{CO}$ or higher combustion efficiency.

We do acknowledge that it may be easier to interpret temporal changes in combustion characteristics for these less-populated but industry-centered cities (e.g. Zibo or Baotou), than for megacties with diverse sectoral signals (e.g., Shanghai or LA).

Here are the revised discussions on Zibo:

| City | Total power capacity (MW) and by fuel types | Key industry OR annual crude steel capacity (kt yr$^{-1}$) |
| --- | --- | --- |
| Los Angeles | 5,808 MW (95.6% fueled by gas; 0% by coal) | refinery, shipping |
| Shanghai | 16,031 MW (75.2% fueled by coal; 24.4% by gas) | iron & steel (25,099 kt yr$^{-1}$) |
| Baotou | 6,470 MW (100% fueled by coal) | iron & steel (12,619 kt yr$^{-1}$) |
| Zibo (w/ Zouping) | 9,720 MW (100% fueled by coal) | electrolytic aluminum; iron & steel (2,532 kt yr$^{-1}$) |

**Table 1.** A summary of total power generation capacity (Global Power Plant Dateset by World Resources Institute, 2018) and information on heavy industry including annual crude iron capacity (GID, Wang et al., 2019). Power plants are selected from a 0.5° × 0.5 °region around each city with percentage generated by the main fuel types.

400    Zibo with the nearby county-level city Zouping accounts for over 1/8 of the total coal consumption of Shandong province in 2017. The coal-fired power plants in the area contribute to a total power generation capacity of 9,720 MW (**Table 1**), which are likely to support local metal industries especially the production of electrolytic aluminum (world's top producers). The maximum $XCO_2$ enhancement per OCO-3 sounding can even reach up to 10 ppm for a few overpasses (not shown). Interestingly, $ER_{CO}$ for Zibo first declined from 10.1±1.1 mmol mol$^{-1}$ to 6.1±0.6 mmol mol$^{-1}$ during Feb 2020 and gradually

405    increased back to 18.2±1.1 mmol mol$^{-1}$ by June 2020 (**Fig. 6b**). Such temporal variations in $ER_{CO}$ agree nicely with the timing of the initial phase of COVID-19 lockdown in China (i.e., Feb to May in 2020) (e.g., Laughner et al., 2021). We suspect changes in $ER_{CO}$ could be driven by the partial shut-down and re-opening of the multiple coal-fired power plants and metal industries in the area.

392-394: "In contrast to LA, ...". It is suggested here that wind speeds and direction do not change dramatically for LA, while the opposite appears to be suggested in line 386.

Lines 392-394 and line 386 may appear to be conflicting but they were referring to two different aspects regarding wind conditions. To clarify, the statement of "relatively small wind changes over LA compared to Shanghai" around lines 392-394 was referring to the changes in wind vectors across overpass dates/seasons (e.g., Feb, April, Dec). While "the dramatic wind changes over LA" around line 386 described the changes in wind vectors between OCO and TROPOMI overpass times but for one overpass. To avoid confusion, we now modified both sentences.

**Line 392-394**: "In contrast to LA, where urban plumes are usually well-constrained with the basin, wind speeds and directions vary across different overpasses over Shanghai —i.e., southeasterly wind on Feb 4 and Feb 20, 2020; southwesterly wind on Feb 24, 2020, and Feb 19, 2021; and northerly wind on April 23 and Dec 30, 2020.

**Line 386**: Although OCO-3 has sampled the Los Angeles basin dozens of times to date, many overpasses did not pass the quality check (i.e., QF) and were removed from the final result due to their noticeable shifts in urban plumes between two overpass times (e.g., March 3, April 15, and May 5, 2020 for LA; discussed in Sect. 4.1).

425: This is an important point for interpreting the results in terms of "combustion efficiency". While in many industrial processes, hydrocarbons are combusted with the intention of achieving as high as possible combustion efficiencies, in iron/steel production, CO actually serves an important function during the process. While an iron/steel plant will aim at burning as much as possible the CO before release to the atmosphere, it still is a tricky comparison with other sectors under the concept of 'combustion efficiency'. Perhaps a short mention of this would be appropriate.

We agree that some explanations of different industrial activities are needed. Here is the paragraph on acknowledging the uniqueness in iron/steel production, compared to other industrial processes:

"We acknowledged that although many iron/steel plants may aim at combusting as much CO as possible before releasing CO into the atmosphere, the indispensable role CO played in the iron/steel industry makes it unique when assessing its $ER_{CO}$ and combustion efficiency among various industrial processes."

**4 Discussion**

459-460: "...constraint can be relaxed". This statement may need to be reconsidered when also taking into account the changes in emissions of (FF) CO2 and CO during the day (see major comments).

We agree with the reviewer and have modified the text as follows:

"If TROPOMI pixel sizes are relatively large (i.e., non-nadir observations) or the wind is steadier, this dt constraint may be relaxed, as long as emissions for a specific city is less driven by sectors with noticeable diurnal cycle (e.g., road transportation)."

461-462: This sentence is not clear to me. The text appears to suggest that the plumes during the overpasses on May 31, 2020 are sufficiently similar. Still, in figure 8 it is marked as an outlier (*). This also makes the relation to the next sentence (461-462) unclear.

We apologize for the confusing tilted label on the x-axis; but if looking closer, the May 31, 2020, one is the second bar from the left and associated with the "0" on the top of the bar. We now adjusted the orientation of these x-labels.

[Figure]

497: I had to reread the paragraph on VOC's several times and I'm still not really sure what is the conclusion on the relevance for the present work. Please consider guiding the reader a bit more.

We have now rewritten the paragraph as:

Another factor that we did not explicitly account for is the secondary CO production from both anthropogenic and biogenic VOCs (AVOCs, BVOCs). Under a cascade of reactions in favorable conditions, VOCs emitted from the upwind source location are oxidized to CO at various rates, which result in possible higher CO at the downwind sounding location and a divergence

530 between enhancement ratios and emission ratios. As BVOCs are usually associated with shorter lifetimes compared to many AVOCs (e.g., Surl et al., 2018), we discuss BVOCs and AVOCs separately. BVOCs can contribute significantly to the total CO source at the regional scale especially during growing seasons (e.g., Miller et al., 2008; Hudman et al., 2008; Gonzalez et al., 2021). However, since BVOCs like biogenic $CO_2$ come mainly from rural areas outside the city, by subtracting localized CO background using CO observations outside the urban plume, the impact from BVOCs on the derivation of CO enhancements

535 would be minimized. The lifetime of most AVOCs remains long enough, except for a few species including alkenes (Surl et al., 2018). Without a good observational constraint of the VOC composition and group-specific emissions for different cities around the globe over the years, it would be challenging to accurately quantify the impact on atmospheric XCO and $ER_{CO}$ due to AVOCs emitted from urban areas or specifically from industrial areas. More future efforts regarding urban VOCs may include 1) exploring what good proxies can be measured from space that well represent the bulk AVOC characteristics (e.g.,

540 formaldehyde, Zhu et al., 2014) and 2) interpreting such observations, e.g., by utilizing chemical transport models for source attribution (Gonzalez et al., 2021). Note that the noise/uncertainty in current daily TROPOMI formaldehyde observations may be too large for daily resolved analyses.

516: "using the industry emissions from EDGAR". Does this comparison indeed only consider industry emissions or also the other sectors (e.g. those shown in Fig S11d-f)?

To clarify, we performed two separate sets of simulations, 1) using total emissions from sectors related to fossil fuel burning from EDGAR (in **Fig. S11b**), and 2) using emissions from separate sectors from

EDGAR (in **Fig. S11d-f**). For the second set of simulations, we have only shown the modeled enhancements from the four major emission sectors, namely 'industry', 'on-road', 'power', and 'resident' as labeled in the initial **Fig S11d-f**. Several non-FF sectors (e.g., AGS - agricultural soils; SWD - solid waste; AWB - agricultural waste burning; and FFF - fossil fuel fires) had not been chosen.

Below are the FF sub-sectors from EDGAR we selected and how we modified them.
**ENE: Power industry **-> renamed as 'power' (Fig. S11e)**
**TRO: Road transportation **-> renamed as 'on-road' (Fig. S11d)**
**RCO: Energy for buildings **-> renamed as 'resident' (Fig. S11f)**

Subsectors that were **combined into one sector as "industry" (Fig. S11c)**
**IND: Combustion for manufacturing**
**CHE: Chemical processes**
**NMM: Non-metallic minerals production**
**NFE: Non-ferrous metals production**
**IRO: Iron and steel production**

The following sub-sectors are associated with very small emissions for our cities and were not shown in Fig S11 previously.
**REF_TRF: Oil refineries and Transformation industry**
**PRO: Fuel exploitation**
**TNR: Other transportation sources**
**NEU: Non-energy use of fuels**
**PRU_SOL: Solvents and products use**

We have now revised the text for all panels for clarifications:

[Figure]

Modeled enhancement ratio [ppb ppm-1] (e.g., XCO_sect / XCO2_sect) using EDGAR-based sector-specific CO or CO2 emissions

[Figure]

**Supplementary information**

Figure S7: The figure does not show the overpass time, that could for example explain the difference between the lower right figures (Zibo 2020-05-05 and Zibo 2020-06-21). Perhaps the colour can be used to indicate overpass time instead of the month (which is already shown above each graph).

We agree that adding overpass time is a good idea because overpass hours and seasons determine the net positive or negative anomalies from the nearby biosphere. We have now modified **Fig. S7** and its associated caption by including the solar zenith angle (SZA) as an indicator for biogenic contributions (see below). For example, for Zibo, contributions from the biosphere are almost all positive on May 05, 2020, at 23 UTC (= 7 am local time) while contributions are almost all negative on June 21, 2020, at 05 UTC (1 pm local time). More generally, biogenic anomalies are more positive when overpass times are in early mornings (high SZA) or in wintertime, due to accumulative influences from net positive biospheric fluxes.

Histogram of modeled biogenic XCO2 anomalies [ppm] per OCO sounding (as a function of SZA [degrees])
Site & overpass time in UTC per panel

**Technical corrections:**

**Abstract**

3: Consider using "combustion efficiency" consistently instead of "burning efficiency".

We have now replaced all "burning efficiency" with "combustion efficiency" throughout the manuscript.

**1 Introduction**

43-45: "Benefit ... reported". Please check the sentence structure, it is not fully clear.

We agree that this sentence sounded awkward and have reworded it:

"The commonly used approach in estimating combustion efficiency is to combine atmospheric observations of multiple trace gases and report the ratio of the total or excess measured concentrations (above a defined background value) between tracers (Silva and Arellano, 2017; Reuter et al., 2019; Park et al., 2021). Such tracer-to-tracer ratio calculation has the benefit that errors in describing the atmospheric transport that carries tracers to the measurement site can be cancelled.

50: "be difficult." some words are missing. Perhaps "...be difficult to identify"?

56: "diagnosed". consider using a different term, e.g. "analysed".

All corrected.

Figure 1:

- What sector(s) is "biofuel + ff" referring to? All anthropogenic combustion sectors?
- In fig.1b the vertical (minor) grid lines fall in between years. The figure may be easier to read if these lines coincide with the years.

Indeed, "biofuel + fossil fuels" is referring to all possible anthropogenic combustion sectors, thereby neither for a specific sector nor for a city. We simply removed those values from Fig. 1:

[Figure]

65-66: Not clear what is meant by "...the generalization and representation of gridded ERs." Also, the next few sentences could be written a bit more clearly.

To clarify, we were trying to state that large variation in ERs makes it difficult to map ERs for the entire globe. For example, adopting one universal number per sector may even be insufficient to account for the contrast in ERs across regions or years. The reason we brought it up is that most gridded emission inventories involve prior assumptions of emission factors. The relevant text has now been rewritten as:

> When estimating fossil fuel emissions from a bottom-up perspective, most inventories rely on activity data and may involve prior knowledge of emission factors (Gurney et al., 2019; Solazzo et al., 2021). One notable example is Hestia, a high-resolution
> 70   inventory for the US, which estimates $CO_2$ emissions of non-point sources based up CO emissions from the National Emission Inventory with EFs and carefully evaluates their adopted EFs (Gurney et al., 2019). However, when constructing emission inventories across regions/nations, the large variability in ERs across combustion processes, sectors, years, and regions (as seen in **Fig. 1a**) makes the choice of EFs extremely challenging. Accurate bottom-up emission estimates require accurate activity data and $EF_X$ that naturally vary with combustion conditions (e.g., temperature, fuel load, oxygen level) and are
> 75   generally not well known especially over data-scarce regions. To our knowledge, only a few global inventories, such as the Emissions Database for Global Atmospheric Research (EDGAR, Solazzo et al., 2021), offer global anthropogenic CO and $CO_2$ emissions. Considering the challenge in approximating ERs, certain knowledge derived from atmospheric observations may 1) complement inventory-based ERs (e.g., $CO:NO_x$ ratio in Lama et al., 2020) and 2) facilitate the emission constraint for a desired gas usually with relatively larger uncertainties (Wunch et al., 2009; Palmer et al., 2006; Wang et al., 2009; Brioude
> 80   et al., 2012; Nathan et al., 2018). Such prior achievements motivate us to examine ERs using satellite observations of multiple tracers.

**2   Data and methodology**

107: "and surrounded coal-fired" —> "surrounding"? or "presence of multiple coal-fired power plants"
109: "that is the spatial extent" —> "that is in the spatial extent"
111: "that estimated" —> "that are estimated"
*111: "FF enhancements" this term is a bit vague.*
116: "we illustrate how much ERs ... can be extracted." What does this mean?
124: "north hemisphere". This term appears to be more commonly spelled as "northern hemisphere".
163: "interfere the calculation" —> "interfere with the calculation"
218: "swaths of stretching" —> "swaths stretching"
220: "have been emphasized" —> "has been emphasized"

Lines 107-220 have been corrected as suggested by the reviewer. For the definition of "FF enhancements", we added a note for clarification:
> "Since we do not differentiate emission signals due to biofuel or fossil fuel (FF) combustion, the term "FF enhancements" is simply referred to *column enhancements induced by any anthropogenic combustion processes* from the target city."

**3   Results**

375: No need to present two decimals for the crude iron production.
Corrected.

**4  Discussion**

451: "considered  when"

456: "Interfering the" —> interfering with the".

456-457: *Please rephrase, as the lack of concurrence is not itself inducing the wind directional shift.*

464: "future geostationary satellite" singular or plural?

464: What is meant by "spontaneously" here?

468: "a urban" —> "an urban"

512: "efficiency can" —> "efficiency, which can"

513-514, 521: "inventory" —> "inventories"

*Section 4.3. the readability of the text in this section could be improved by a thorough read through*

> All corrected. Line 456-457 is now reworded as "The biggest challenge affecting the robust spatially-resolved ER estimates is the shift in wind directions across different satellite overpass times." We have revised the text (please refer to **Sect. 4.3 and 4.4** in the revised version for more details).

**5  Conclusion**

545: "industry-related ER_CO slightly lower"—> "industry-related ER_CO are slightly lower"

> Corrected.

**Supplementary information**

Figure S10: "distribution is stewed towards" —> "distribution is skewed towards"

> Corrected. We truly thank the reviewer for the careful read-through!

---

## Referee Report (RR1)

**Wu et al., 2022: Towards sector-based attribution using intra-city variations in satellite-based emission ratios between CO2 and CO. ACP**

**Review of revised submission**

The authors have provided thorough, point-by-point responses to the questions and comments raised by this reviewer. In the revised submission, improvements have been carefully implemented through updated text, several updated figures and even additional data collection and analysis.

In particular additional data collection and analysis has been performed to address the major comment regarding the potential effect of specific daily temporal patterns followed by some sectors, such as road transport and the residential sector, leading to changing ER_CO during the day, which could present an additional source of mismatch between observed XCO and XCO2 even when wind conditions remain relatively constant. By using TCCON data of simultaneously retrieved CO and CO2 signals, collected near LA, the authors show that, although the ER_CO may change substantially during the day, for the overpasses that went into the final result for LA, the differences were small. As the potential mismatch is now clearly discussed in the text, this comment can be considered resolved.

The authors also addressed the remaining minor and technical comments and generally improved the clarity and readability of the manuscript, resulting in a high-quality paper.